# Pyruvate and related energetic metabolites modulate resilience against high genetic risk for glaucoma

Keva Li[1]*[†], Nicholas Tolman[2][†], Ayellet V Segrè[3,4], Kelsey V Stuart[5], Oana A Zeleznik[6], Neeru A Vallabh[7,8], Kuang Hu[5], Nazlee Zebardast[3], Akiko Hanyuda[9,10], Yoshihiko Raita[11], Christa Montgomery[2], Chi Zhang[2], Pirro G Hysi[12,13], Ron Do[14], Anthony P Khawaja[5], Janey L Wiggs[3,4], Jae H Kang[6][‡], Simon WM John[2,15]*[‡], Louis R Pasquale[1]*[‡], UK Biobank Eye and Vision Consortium

[1]Department of Ophthalmology, Icahn School of Medicine at Mount Sinai, New York, United States; [2]Department of Ophthalmology, Vagelos College of Physicians and Surgeons, Columbia University Irving Medical Center, New York, United States; [3]Department of Ophthalmology, Massachusetts Eye and Ear, Harvard Medical School, Boston, United States; [4]Broad Institute of MIT and Harvard, Cambridge, United States; [5]NIHR Biomedical Research Centre, Moorfields Eye Hospital NHS Foundation Trust, and University College London Institute of Ophthalmology, London, United Kingdom; [6]Channing Division of Network Medicine, Department of Medicine, Harvard Medical School and Brigham and Women's Hospital, Boston, United States; [7]Department of Eye and Vision Science, Institute of Life Course and Medical Sciences, University of Liverpool, Liverpool, United Kingdom; [8]St. Paul's Eye Unit, Liverpool University Hospital NHS Foundation Trust, Liverpool, United Kingdom; [9]Department of Ophthalmology, Keio University School of Medicine, Tokyo, Japan; [10]Epidemiology and Prevention Group, Center for Public Health Sciences, National Cancer Center, Tokyo, Japan; [11]Okinawa Kenritsu, Chubu Byoin, Uruma, Okinawa, Japan; [12]Department of Ophthalmology, St Thomas' Hospital, King's College London, London, United Kingdom; [13]Department of Twin Research & Genetic Epidemiology, St Thomas' Hospital, King's College London, London, United Kingdom; [14]Department of Genetics and Genomics Science, Icahn School of Medicine at Mount Sinai, New York, United States; [15]Zuckerman Mind Brain Behavior Institute, Columbia University, New York, United States

*For correspondence:
keva.li@icahn.mssm.edu (KL);
sj2967@cumc.columbia.edu
(SWMJ);
louis.pasquale@gmail.com (LRP)

[†]These authors contributed equally to this work
[‡]These authors also contributed equally to this work

## eLife Assessment

This study presents a **valuable** finding on the importance of the plasma metabolome in glaucoma risk prediction. The evidence supporting the claims of the authors is **solid** and the work offers insights for the design of protective therapeutic strategies for glaucoma. The authors have addressed the concerns of the reviewers and reported on the limitations of the study.

**Abstract** A glaucoma polygenic risk score (PRS) can effectively identify disease risk, but some individuals with high PRS do not develop glaucoma. Factors contributing to this resilience remain unclear. Using 4,658 glaucoma cases and 113,040 controls in a cross-sectional study of the UK Biobank, we investigated whether plasma metabolites enhanced glaucoma prediction and if a metabolomic signature of resilience in high-genetic-risk individuals existed. Logistic regression

models incorporating 168 NMR-based metabolites into PRS-based glaucoma assessments were developed, with multiple comparison corrections applied. While metabolites weakly predicted glaucoma (Area Under the Curve = 0.579), they offered marginal prediction improvement in PRS-only-based models (p=0.004). We identified a metabolomic signature associated with resilience in the top glaucoma PRS decile, with elevated glycolysis-related metabolites—lactate (p=8.8E-12), pyruvate (p=1.9E-10), and citrate (p=0.02)—linked to reduced glaucoma prevalence. These metabolites combined significantly modified the PRS-glaucoma relationship ($P_{interaction}$ = 0.011). Higher total resilience metabolite levels within the highest PRS quartile corresponded to lower glaucoma prevalence (Odds Ratio$_{highest\ vs.\ lowest\ total\ resilience\ metabolite\ quartile}$=0.71, 95% Confidence Interval = 0.64–0.80). As pyruvate is a foundational metabolite linking glycolysis to tricarboxylic acid cycle metabolism and ATP generation, we pursued experimental validation for this putative resilience biomarker in a human-relevant *Mus musculus* glaucoma model. Dietary pyruvate mitigated elevated intraocular pressure (p=0.002) and optic nerve damage (p<0.0003) in *Lmx1b*$^{V265D}$ mice. These findings highlight the protective role of pyruvate-related metabolism against glaucoma and suggest potential avenues for therapeutic intervention.

## Introduction

Glaucoma is a polygenic, progressive neurodegenerative disease and a leading cause of irreversible blindness (*Soh et al., 2021*). The disease is typically asymptomatic until advanced visual field loss occurs, and around 50 to 70% of people affected remain undiagnosed (*Khawaja and Viswanathan, 2018*; *Jan et al., 2024*). Early detection and intervention are essential to stop disease progression and prevent visual impairment in glaucoma-affected individuals, as there is no cure available; however, population-based glaucoma screening is not cost-effective from a public health perspective (*Moyer, 2013*; *US Preventive Services Task Force, 2022*; *Zukerman et al., 2021*).

Glaucoma is well-suited for developing and applying a PRS to facilitate disease identification and risk stratification as it is a condition with several endophenotypes, such as elevated intraocular pressure (IOP) and retinal nerve fiber layer (RNFL) thinning, and it is highly heritable (*Craig et al., 2020*; *de Vries et al., 2025*; *Sekimitsu et al., 2023*). Glaucoma risk is influenced by both genetic and metabolic factors, with emerging evidence suggesting that gene-environment interactions may play a greater role in conferring disease risk than independent exposures alone (*Stuart et al., 2023a*; *Kang et al., 2016a*; *Kang et al., 2018*; *Kang et al., 2016b*; *Stuart et al., 2023b*; *Doucette et al., 2015*).

Glaucoma was five times more likely in the UK Biobank (UKBB) among participants with the highest genetic risk decile versus the lowest decile (7.4 vs 1.3%) (*Sekimitsu et al., 2023*). While the relatively low glaucoma prevalence in the highest decile group could be explained by disease under-ascertainment or a PRS that incompletely reflects the glaucoma genetic architecture, it is possible that resilience metabolite biomarkers could explain this result, in addition to protective genetic or epigenetic factors.

Recent advancements in metabolomics have opened up avenues to explore metabolites as potential biomarkers for glaucoma (*Nazifova-Tasinova et al., 2020*). Metabolites are intermediate and end products of cellular processes critical for driving cellular growth and tissue homeostasis (*Johnson et al., 2016*). These small molecules provide a holistic measure of physiological status, reflecting both genetic predispositions and environmental influences. Previous studies have indicated a potential role for plasma metabolites to stratify glaucoma risk; however, these studies were limited by small sample sizes (*Javadiyan et al., 2012*; *Burgess et al., 2015*; *Kouassi Nzoughet et al., 2020*; *Leruez et al., 2018*; *Wang et al., 2021*), with restricted coverage of metabolites. In addition, the potential benefits of the integration of metabolomics with genetics to identify individuals at the highest risk of glaucoma remain unexplored.

This study aims to evaluate plasma metabolites as risk factors for glaucoma. First, we explored whether incorporating plasma metabolite data improved the predictive accuracy of a PRS for glaucoma risk based on genetic and metabolomic data available for 117,698 UKBB participants. Additionally, we evaluated interactions between glaucoma PRS and metabolomics for further stratifying individuals with high genetic risk but without glaucoma. We undertook an agnostic approach to identify a metabolite signature associated with resilience to high glaucoma genetic risk. Finally, we experimentally

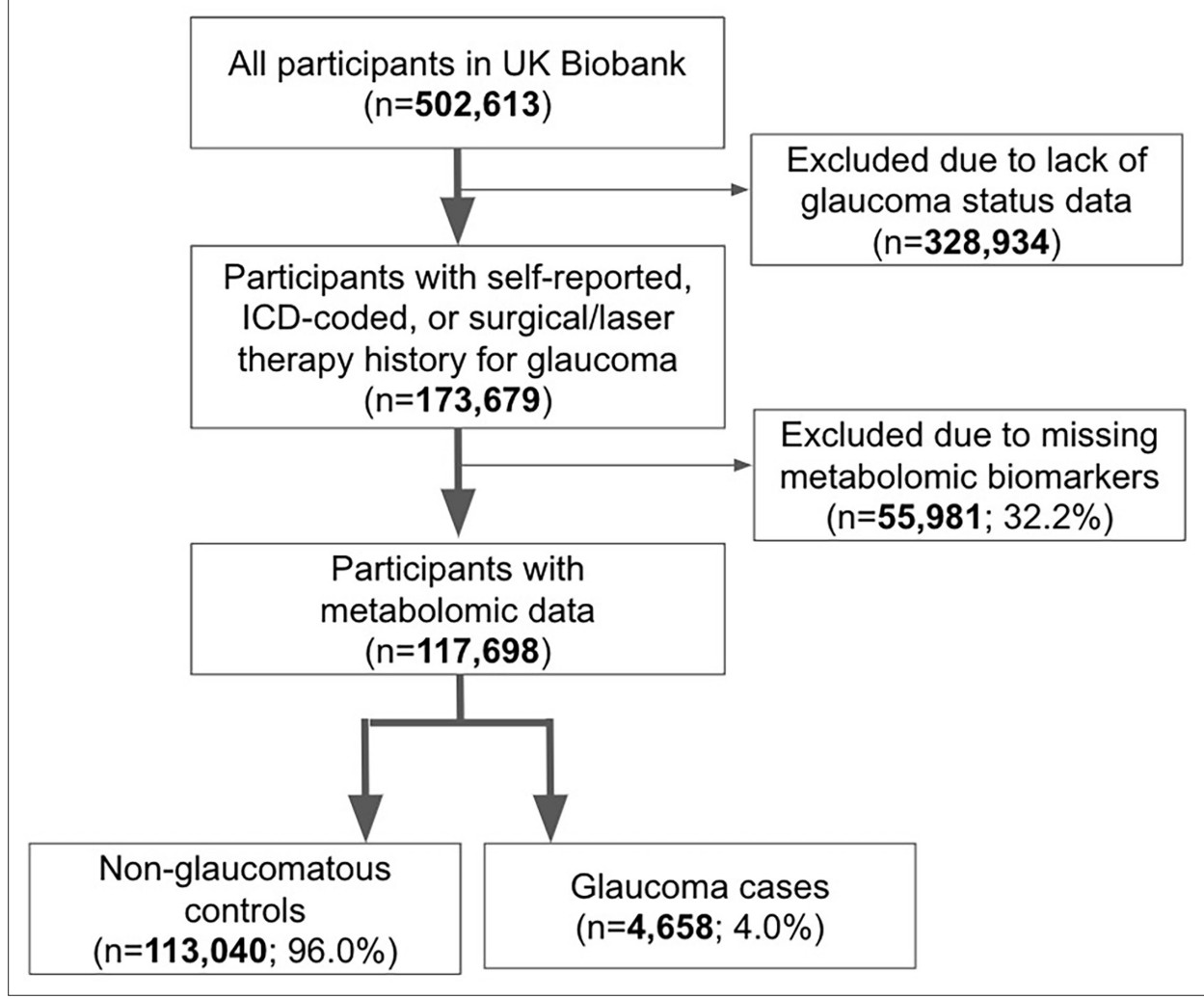

**Figure 1.** Participant flow chart describing inclusion and exclusion criteria from the UK Biobank.

validated a top resilience metabolite by assessing its ability to rescue the ocular phenotype in a human-relevant, genetic mouse model of glaucoma.

## Results

### UK Biobank study characteristics

A total of 117,698 participants (4,658 glaucoma cases and 113,040 non-cases) were included in this study. UKBB participants were predominately of European ancestry (85.8%) but were also of mixed American (0.22%), Asian (2.77%), and African (1.73%) ancestry. The design of the human studies is depicted graphically in *Figure 1* and *Figure 2*. There were significant differences seen in baseline demographic and clinical characteristics between the glaucoma cases and the non-cases presented in *Table 1*. Notably, characteristics associated with glaucoma included male gender, older age, genetic African ancestry (see *Privé et al., 2022* for the method to determine genetically inferred ancestry in the UKBB), prior history of smoking, slightly lower cholesterol levels, higher body mass index (BMI), higher hemoglobin A1c (HbA1c), higher IOP, thinner macular retinal nerve fiber layer (mRNFL) thickness, higher caffeine intake, higher alcohol intake, higher oral steroid use, diabetes, and coronary artery disease.

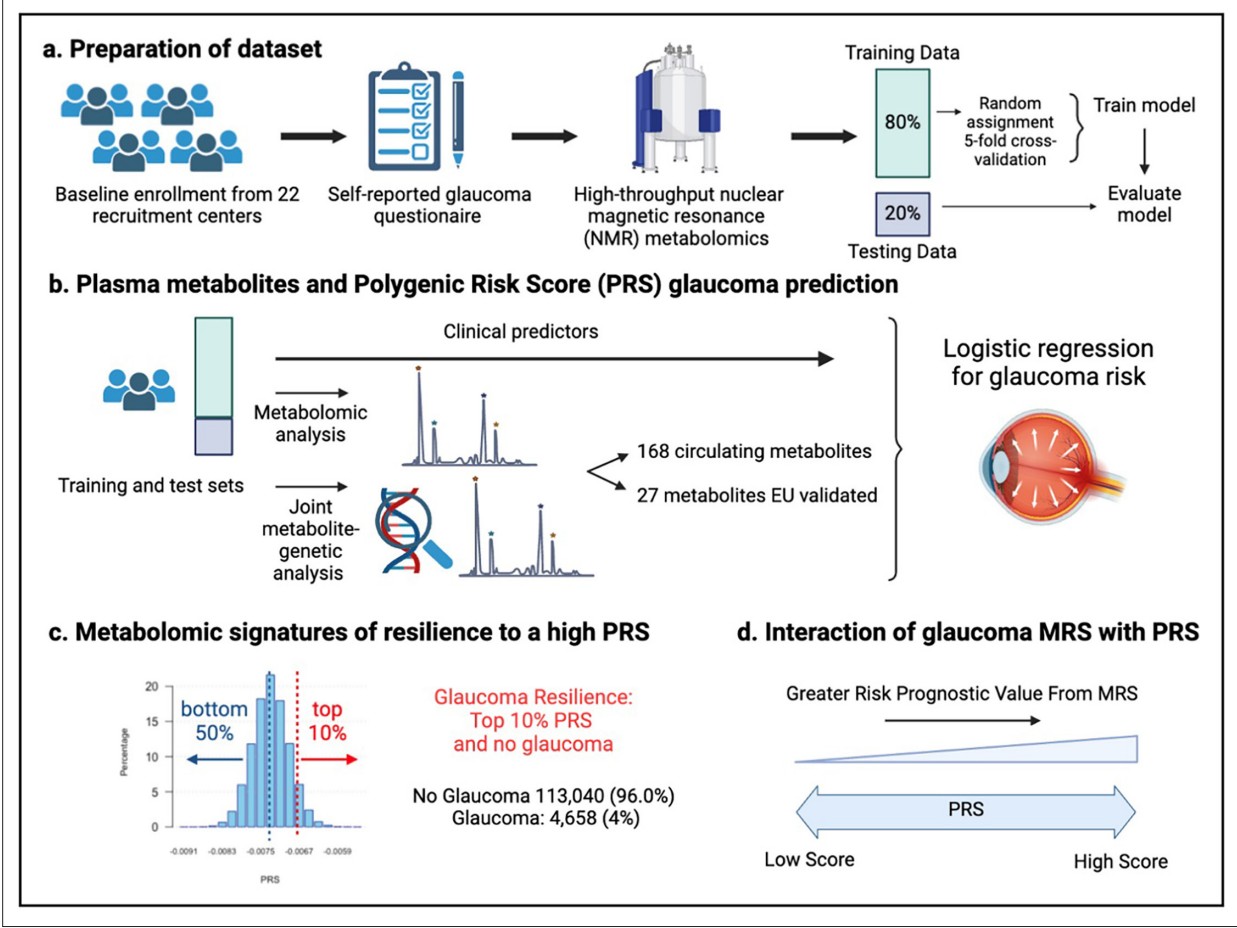

**Figure 2.** Study design from the UK Biobank. (**a**) 117,698 individuals had metabolomics data available from the UK Biobank, which was divided into a training and test set to formulate a metabolic risk score (MRS) model. (**b**) The inclusion of metabolites (either 168 metabolites on the nuclear magnetic resonance (NMR) platform or a subset of 27 metabolites with European Union (EU) certification) in relation to prevalent glaucoma risk prediction was studied. (**c**) A histogram of the polygenic risk score (PRS) distribution is shown. Overall, 4,658 cases and 113,040 individuals without glaucoma are available for analysis. The metabolomic signature of resilience to the top 10% of glaucoma PRS was assessed among 1,693 cases (14.4%) and 10,077 individuals without glaucoma (85.6%). (**d**) Interactions of prevalent glaucoma with MRS and PRS quartiles were examined.

## Plasma metabolites marginally improve glaucoma risk prediction

We first sought to determine whether the addition of metabolites could improve the prediction of glaucoma from basic demographics, clinical variables, and genetic data. We created four logistic regression models, each with increasing predictive variables considered, to evaluate their added utility for predicting glaucoma (see Methods and *Figure 3—source data 1* for model construction details). We analyzed two sets of metabolite variables: a comprehensive set of 168 metabolites measured by nuclear magnetic resonance (NMR) spectroscopy included in the UKBB dataset and a limited set of 27 metabolites approved by the European Union (EU) for in vitro diagnostic use.

To evaluate the performance of each model, we plotted receiver operating characteristic (ROC) curves and calculated the area under the curve (AUC) metrics. In model 1, which utilized only metabolite data, the highest performance was achieved using the full panel of 168 metabolites, yielding an AUC value of 0.602 (95% CI = 0.592–0.612) compared to 0.579 (95% CI = 0.569–0.589; p=0.0003) using the 27 metabolites (*Figure 3*). Model 2, which incorporated demographic information, and model 3, which also included clinical variables, exhibited similar performance before the addition of metabolite data. In model 2, the addition of the 168 metabolite panel demonstrated the best performance, with an AUC value of 0.670 (95% CI = 0.660–0.680) compared to 0.664 (95% CI = 0.654–0.674) for the model without metabolites (p=0.002) and 0.666 (95% CI = 0.656–0.676) for the limited panel of 27 metabolites. This performance trend persisted in model 3 with the inclusion of 168 metabolites,

**Table 1.** Demographic and clinical characteristics of the UK Biobank study population assessed in 2006–2010.

| Characteristic | No Glaucoma | Glaucoma | p-value |
|---|---|---|---|
| Sample Size (%) | 113,040 (96) | 4,658 (4) | |
| Sex - Male (%) | 52,497 (46.4) | 2,493 (53.5) | <0.001 |
| Age at recruitment, years, (mean (SD)) | 56.7 (8.0) | 60.9 (6.6) | <0.001 |
| Ethnicity (%) | | | 0.0010 |
| White | 105,912 (94) | 4,331 (93) | |
| Asian | 2,977 (2.6) | 127 (2.7) | |
| Black | 2,228 (2.0) | 130 (2.8) | |
| Other | 1,923 (1.7) | 70 (1.5) | |
| Genetic Ancestry (%) | | | <0.001 |
| African | 1,911 (1.7) | 123 (2.7) | |
| AMR | 261 (0.2) | 2 (0.0) | |
| Asian | 3,125 (2.8) | 134 (2.9) | |
| European | 96,991 (86.6) | 4,001 (86.7) | |
| Smoking Status (%) | | | <0.001 |
| Never | 63,240 (56) | 2,339 (52) | |
| Prefer not to answer | 430 (0.4) | 16 (0.4) | |
| Previous | 38,926 (34) | 1,732 (39) | |
| Current | 10,605 (9.4) | 410 (9.1) | |
| Total Cholesterol, mmol/l (median [IQR]) | 4.6 [3.98, 5.22] | 4.5 [3.88, 5.18] | <0.001 |
| Physical Activity, MET-minutes per week (mean (SD)) | 2,465 (2,430) | 2,458 (2,438) | 0.84 |
| Body Mass Index, kg/m$^2$ (mean (SD)) | 27.4 (4.8) | 27.7 (4.7) | <0.001 |
| HbA1c, mmol/mol (mean (SD)) | 36.0 (5.8) | 37.4 (6.8) | <0.001 |
| Spherical Equivalent, diopter (mean (SD)) | –0.1 (2.1) | –0.1 (1.9) | 0.67 |
| Intraocular pressure, mmHg (mean (SD)) | 15.9 (2.6) | 17.8 (4.5) | <0.001 |
| mRNFL thickness, µm (mean (SD)) | 28.7 (1.8) | 28.5 (1.3) | <0.001 |
| Beta blocker use (%) | 8398 (7.4) | 447 (9.6) | <0.001 |
| Caffeine intake, mg/day (mean (SD)) | 165 (67) | 168 (59) | 0.0020 |
| Alcohol intake, g/week (median [IQR]) | 84 [40.3, 146.9] | 84 [48.0, 154.6] | <0.001 |
| Diabetes (%) | 6,512 (5.8) | 463 (9.9) | <0.001 |
| Oral steroid use (%) | 3,043 (2.7) | 184 (4.0) | <0.001 |
| Coronary Artery Disease (%) | 5,153 (4.6) | 377 (8.1) | <0.001 |

SD, standard deviation; IQR, interquartile range; AMR, mixed American; MET metabolic equivalent of task; mRNFL, macular retinal nerve fiber layer.

producing an AUC of 0.680 (95% CI = 0.660–0.700), representing an increase in AUC (p=0.02) from model 3 with the exclusion of metabolites (AUC 0.670; 95% CI = 0.650–0.690). We subsequently examined whether integrating metabolite data into a PRS-based model could improve glaucoma prediction algorithms. In model 4, the panel of 168 metabolites yielded the best performance with an AUC value of 0.806 (95% CI = 0.796–0.816), while the model without metabolites (AUC = 0.801, 95% CI = 0.791–0.811; p=0.004), as well as the model with a limited panel of 27 metabolites (AUC = 0.802, 95% CI = 0.792–0.812; p=0.006) both had lower AUC values. Thus, these findings suggest that

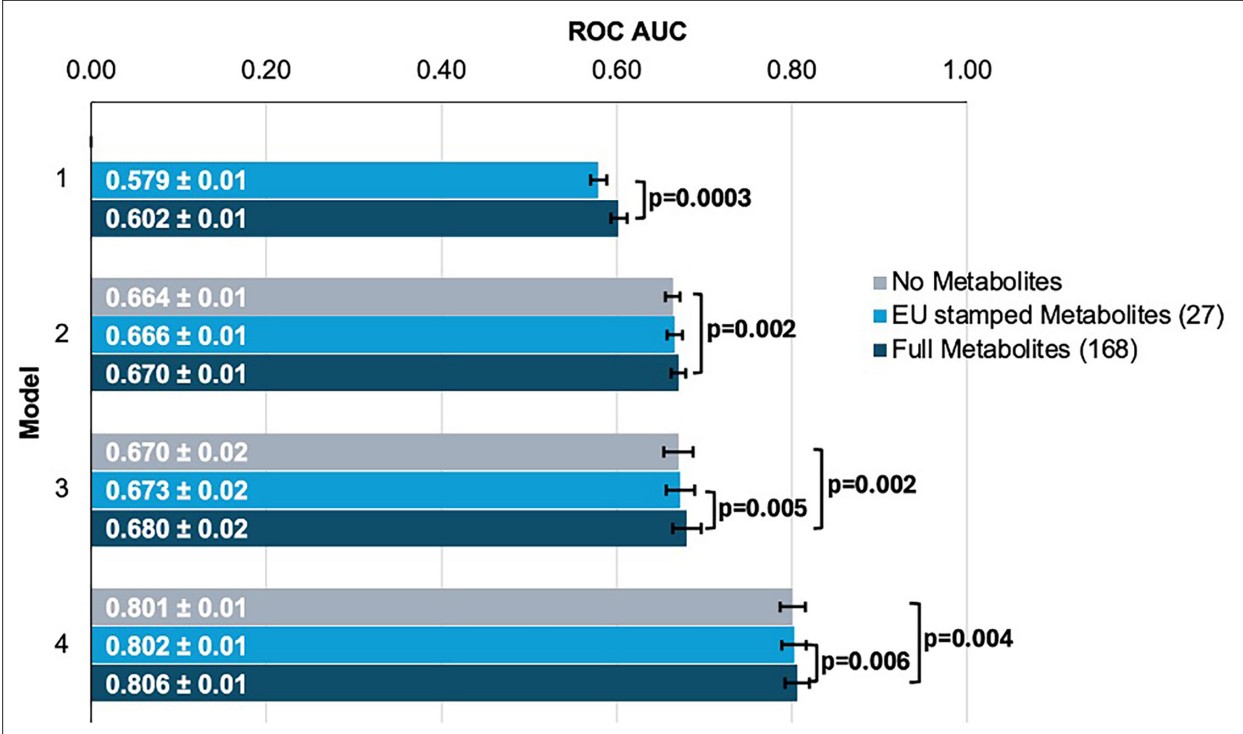

**Figure 3.** Inclusion of metabolite data into glaucoma prediction algorithms. Model 1 includes metabolites only; Model 2 incorporates additional covariates including age (years), sex, genetic ancestry, season, time of day of specimen collection, and fasting time; Model 3 incorporates covariates in Model 2 and smoking status (never, past, and current smoker), alcohol intake (g/week), caffeine intake (mg/day), physical activity (metabolic equivalent of task [MET], hours/week), body mass index (kg/m²), average systolic blood pressure (mm Hg), history of diabetes, HbA1c (mmol/mol), history of coronary artery disease, systemic beta-blocker use, oral steroid use, and spherical equivalent refractive error (diopters); Model 4 incorporates covariates in model 3 and a glaucoma polygenic risk score (PRS). Each color represents a different panel of metabolites (gray = no metabolites; light blue = 27 metabolites; and dark blue = 168 metabolites). The white text represents the AUC ± 95% confidence interval. Abbreviations: ROC, receiver operator curve; AUC, area under the curve; EU, European Union.

The online version of this article includes the following source data for figure 3:

**Source data 1.** Metabolite data beta (effect size) values by model and metabolite groupings.

the addition of metabolite data marginally enhances the prediction of glaucoma beyond the use of demographic and genetic data.

We then investigated whether the inclusion of the 168 metabolites could improve the predictive value for glaucoma by stratifying patients into specific subgroups and calculating AUC curves for each stratum. Accordingly, we found that metabolites showed a modestly improved predictive value for glaucoma among people of White ethnicity (p<0.001), those aged 55 and older (p=0.002), and males (p=0.002) (*Table 2*).

## Metabolites associated with resilience to a high glaucoma polygenic risk score

As metabolite data only marginally augmented clinical or PRS predictions of glaucoma, we hypothesized that metabolites might provide utility for differentiating patients within risk groups. Specifically, we studied participants who possessed a high glaucoma PRS but did not have glaucoma. For this study, we labeled them resilient while recognizing there are many reasons they may not have glaucoma. Participants were stratified into PRS deciles, and metabolite signatures were identified to differentiate patients with and without glaucoma in the top decile (N=11,770) and the bottom half (N=58,358) of the glaucoma PRS distribution (*Figure 4*). Within the top decile of glaucoma PRS, compared to participants without glaucoma, participants with glaucoma were more likely to be older, male, of White ethnicity, and were prior smokers. Glaucoma participants with the highest PRS also had higher BMI, higher HbA1c, higher spherical equivalent, consumed more caffeine and alcohol, and

**Table 2.** Stratification of glaucoma by ethnicity, age, and gender for predictive assessment with and without using metabolite data.

| Stratification | Sample size | Glaucoma cases | No Metabolites (AUC) | Metabolites (AUC) | p-value |
|---|---|---|---|---|---|
| Ethnicity | | | | | |
| White | 110,243 | 4,331 | 0.675 | 0.686 | <0.001 |
| Asian | 3,104 | 127 | 0.780 | 0.768 | 0.062 |
| Black | 2,358 | 130 | 0.704 | 0.706 | 0.52 |
| Age | | | | | |
| <55 years | 43,648 | 788 | 0.576 | 0.566 | 0.52 |
| ≥55 years | 74,050 | 3,870 | 0.569 | 0.596 | 0.002 |
| Gender | | | | | |
| Female | 62,708 | 2,165 | 0.689 | 0.696 | 0.25 |
| Male | 54,990 | 2,493 | 0.659 | 0.673 | 0.002 |

The area under the curve (AUC) of the receiver operating characteristic (ROC) curve was calculated for each demographic stratification to evaluate the predictive performance of models both with and without metabolite data. Differences in model AUC were tested using the DeLong test, and p-values were reported. For models excluding metabolite data, the predictors include as appropriate, age (years), sex, genetic ancestry, season, time of day of specimen collection, fasting time (hours), smoking status (never, past, and current smoker), alcohol intake (g/week), caffeine intake (mg/day), physical activity (metabolic equivalent of task [MET], hours/week), body mass index (kg/m2), average systolic blood pressure (mm Hg), history of diabetes, HbA1c (mmol/mol), history of coronary artery disease, systemic beta-blocker use, oral steroid use, and spherical equivalent refractive error (diopters). Models including metabolite data incorporated the same predictors with the addition of the 168 metabolite measurements.

were more likely to have diabetes and coronary artery disease (*Table 3*). Among participants in the bottom half of glaucoma PRS, participants with glaucoma were also more likely to be older, male, of Black and Asian ethnicity, prior smokers, had higher BMI and higher HbA1c, and were more likely to have diabetes and coronary artery disease (*Table 4*). As expected, participants with glaucoma in both bins of glaucoma genetic risk had higher IOP and thinner mRNFL thickness.

Our multivariable-adjusted analysis revealed that higher probit-transformed levels of lactate (adjusted $P_{NEF}$ = 8.8E-12), pyruvate (adjusted $P_{NEF}$ = 2.9E-10), and citrate (adjusted $P_{NEF}$ = 0.018) were independently associated with no glaucoma in the top decile of glaucoma PRS (*Table 5*). In addition, lower levels of triglycerides and higher levels of selected HDL analytes had an adjusted $P_{NEF}$ <0.2 and were associated with no glaucoma among participants with high genetic risk. Among the bottom half of the PRS distribution, higher levels of albumin were associated with no glaucoma (adjusted $P_{NEF}$ = 0.047), while higher levels of small HDL, omega-3 fatty acids, docosahexaenoic acid, lactate, and citrate were associated with no glaucoma, albeit with an adjusted $P_{NEF}$ <0.2.

## Interaction of metabolic and genetic biomarkers

The three metabolites (lactate, pyruvate, and citrate) for which higher levels were associated with reduced glaucoma prevalence in individuals with high genetic susceptibility had no statistically significant relationship with glaucoma for participants in the lower half of the PRS distribution at the adjusted $P_{NEF}$ <0.05 level. Thus, we hypothesized that there may be an interaction between these metabolites and genetic risk. Such an interaction would suggest that these glycolysis/tricarboxylic acid (TCA) cycle metabolites are relevant primarily in the setting of high PRS, establishing them as resilience factors against elevated genetic risk of glaucoma. Indeed, we observed a significant interaction between elevated levels of total lactate, pyruvate, and citrate with glaucoma PRS quartile for predicting the risk of glaucoma ($P_{interaction}$ = 0.011; *Figure 5*).

To confirm and visualize this interaction, we calculated and plotted the glaucoma odds ratio (OR) as a function of the PRS quartile with each genetic predisposition bin stratified by the sum of the resilience-metabolite probit score quartiles (*Figure 5*). Notably, high levels of resilience-associated metabolites were significantly associated with lower odds of glaucoma, particularly within the higher PRS quartiles (Q3 and Q4). For example, for individuals in PRS Q3 with total resilience-associated

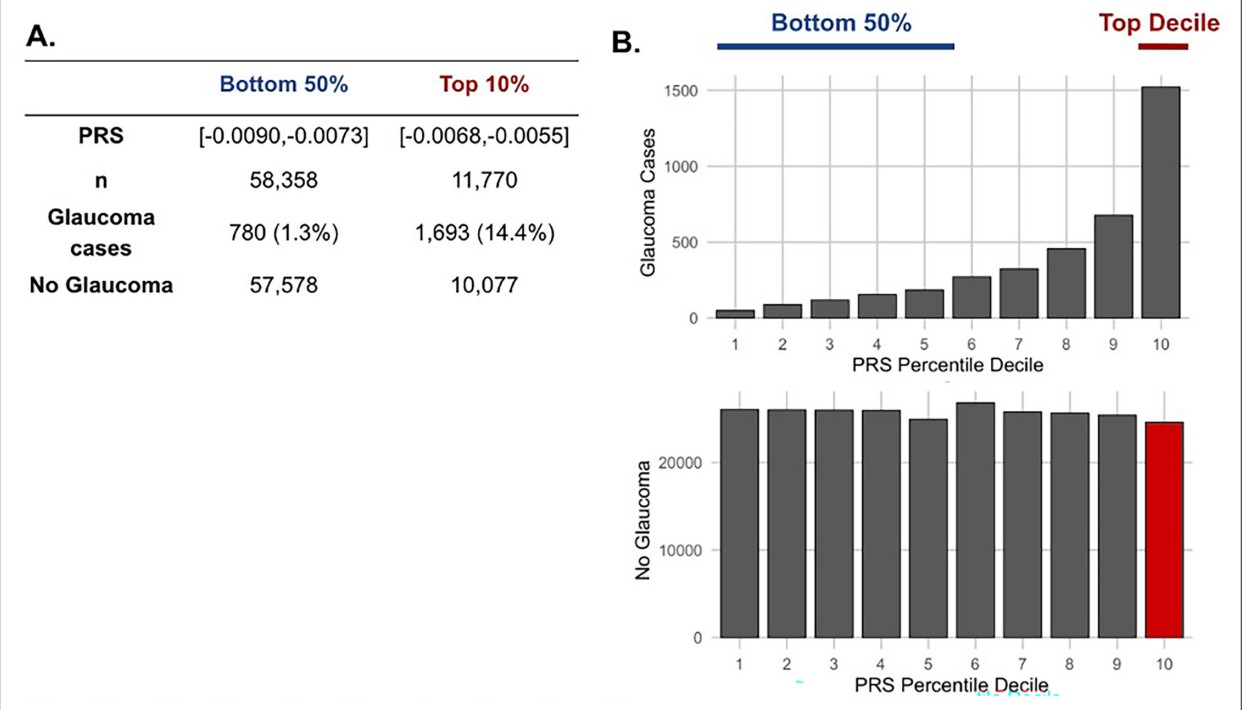

**Figure 4.** The distribution of glaucoma cases and no glaucoma stratified by polygenic risk score (PRS) deciles. Participants were divided into the bottom 50% and top 10% based on their glaucoma PRS, where the prevalence of glaucoma cases from (**A**) the bottom 50% (n=58,358) was 1.3% and from the top 10% (n=11,770) was 14.4%. (**B**) Box plot illustrating the distribution of participants with glaucoma (top) and no glaucoma (bottom) as a function of PRS decile. The blue line denotes participants at the bottom 50% of glaucoma PRS, the red line highlights the participants at the top decile of glaucoma PRS, and the red box represents the participants resilient to glaucoma despite high PRS.

metabolite sum Q3, the glaucoma OR was 0.78 (95% CI = 0.65–0.93; p=0.0059), using metabolite sum Q1 as the reference. In the same PRS Q3 with a higher quartile of total resilience-associated metabolite sum (Q4), the OR was 0.74 (95% CI = 0.62–0.88; p=0.0009). The lowest OR of glaucoma was observed in participants with both the highest PRS (Q4) and highest resilience-associated metabolite sum (Q4), with an OR of 0.71 (95% CI = 0.64–0.80; p<0.001). Although both age and PRS were significant predictors of glaucoma prevalence (both are strongly associated with increased risk), there was no evidence of a significant three-way interaction between the resilience-associated metabolite sum, PRS, and age ($P_{3\text{-way-interaction}}$=0.65).

Next, we investigated whether a holistic metabolic risk score (MRS) incorporating resilience metabolites and other measured metabolites (full panel of 168 metabolites; see Methods and **Appendix 2—table 1** for the beta coefficients used in MRS construction) could be used in conjunction with a glaucoma PRS to improve risk prediction. To quantify the degree of interaction of glaucoma PRS and MRS, we plotted glaucoma OR at various quartiles of glaucoma PRS and MRS compared to the first MRS quartile within each PRS quartile. We found a significant interaction between glaucoma MRS built from 168 metabolites and PRS ($P_{interaction}$ = 0.0012; see **Appendix 2—figure 1A**).

In stratified analyses by quartile of PRS, compared to the first quartile of MRS, higher MRS was associated with increased glaucoma prevalence across most PRS quartiles, with the most pronounced effect observed in individuals in the highest PRS and MRS quartile with an OR of 2.14 compared to those in the highest PRS and lowest MRS quartile (**Appendix 2—figure 1B**). Among individuals in the highest MRS quartile (Q4), PRS Q1 had an OR of 1.66 (95% CI = 1.15–2.42; p=0.0086), PRS Q2 had an OR of 2.08 (95% CI = 1.62–2.67; p<0.001), PRS Q3 had an OR of 2.20 (95% CI = 1.83–2.65; p<0.001), and PRS Q4 had an OR of 2.14 (95% CI = 1.91–2.40; p<0.001). This suggests potential synergistic effects of genetic and metabolite risk factors on glaucoma risk that transcend the impacts of the resilience metabolites.

Most importantly, using individuals in the lowest quartiles of PRS and MRS (PRS Q1, MRS Q1) as the reference group for the entire population (n=117,698), we determined glaucoma OR as a function

**Table 3.** Demographic and clinical characteristics in 2006–2010 of UK Biboank participants among the top 10% of glaucoma polygenic risk score.

| Characteristic | No Glaucoma | Glaucoma | p-value |
|---|---|---|---|
| Sample size (%) | 10,077 (85.6) | 1,693 (14.4) | |
| Sex - Male (%) | 4,667 (46.3) | 893 (52.7) | <0.001 |
| Age at recruitment, years, (mean (SD)) | 56.3 (8.0) | 61.0 (6.3) | <0.001 |
| Ethnicity (%) | | | <0.001 |
| White | 9,256 (91.9) | 1,636 (96.6) | |
| Asian | 279 (2.8) | 24 (1.4) | |
| Black | 345 (3.4) | 20 (1.2) | |
| Other | 197 (2.0) | 13 (0.8) | |
| Smoking Status (%) | | | 0.0030 |
| Never | 5,806 (57.6) | 899 (53.1) | |
| Prefer not to answer | 48 (0.5) | 5 (0.3) | |
| Previous | 3402 (33.8) | 640 (37.8) | |
| Current | 821 (8.1) | 149 (8.8) | |
| Physical Activity, MET-minutes per week (mean (SD)) | 2,439 (2,352) | 2,514 (2,480) | 0.23 |
| Body mass index, kg/m$^2$ (mean (SD)) | 27.4 (4.7) | 27.7 (4.8) | 0.0080 |
| HbA1c, mmol/mol (mean (SD)) | 36.00 (5.9) | 37.0 (6.1) | <0.001 |
| Spherical Equivalent, diopter (mean (SD)) | −0.3 (2.2) | −0.2 (1.9) | 0.0090 |
| Intraocular pressure, mmHg (mean (SD)) | 17.2 (3.0) | 18.3 (5.0) | <0.001 |
| mRNFL thickness, μm (mean (SD)) | 28.7 (1.8) | 28.5 (1.2) | <0.001 |
| Beta blocker use (%) | 719 (7.1) | 179 (10.6) | <0.001 |
| Caffeine intake, mg/day (mean (SD)) | 164 (65) | 170 (60) | 0.0010 |
| Alcohol intake, g/week (median [IQR]) | 84 [38.6, 145.3] | 84 [51.5, 155.3] | 0.002 |
| Diabetes mellitus (%) | 607 (6.0) | 151 (8.9) | <0.001 |
| Oral steroid use (%) | 292 (2.9) | 59 (3.5) | 0.22 |
| Coronary Artery Disease (%) | 443 (4.4) | 148 (8.7) | <0.001 |
| Total cholesterol, mmol/l (median [IQR]) | 4.61 [4.00, 5.25] | 4.54 [3.90, 5.21] | 0.006 |
| Lactate, mmol/l (median [IQR]) | 3.96 [3.27, 4.78] | 3.79 [3.16, 4.58] | <0.001 |
| Pyruvate, mmol/l (median [IQR]) | 0.080 [0.06, 0.10] | 0.077 [0.06, 0.09] | <0.001 |
| Citrate, mmol/l (median [IQR]) | 0.065 [0.06, 0.07] | 0.065 [0.06, 0.07] | 0.87 |
| Cholesteryl Esters in Small HDL, mmol/l (median [IQR]) | 0.33 [0.30, 0.36] | 0.33 [0.30, 0.36] | 0.005 |
| Triglycerides in Very Large VLDL, mmol/l (median [IQR]) | 0.091 [0.05, 0.15] | 0.10 [0.05, 0.17] | <0.001 |
| Alanine, mmol/l (median [IQR]) | 0.29 [0.24, 0.35] | 0.29 [0.24, 0.35] | 0.58 |
| Triglycerides in Chylomicrons and extremely Large VLDL, mmol/l (median [IQR]) | 0.088 [0.03, 0.18] | 0.10 [0.04, 0.21] | <0.001 |
| Acetoacetate, mmol/l (median [IQR]) | 0.010 [0.01, 0.02] | 0.011 [0.01, 0.02] | <0.001 |
| Cholesteryl Esters in Medium HDL, mmol/l (median [IQR]) | 0.41 [0.34, 0.48] | 0.40 [0.34, 0.47] | 0.003 |
| Triglycerides in Large VLDL, mmol/l (median [IQR]) | 0.15 [0.09, 0.22] | 0.16 [0.10, 0.23] | <0.001 |
| Cholesterol in Medium HDL, mmol/l (median [IQR]) | 0.49 [0.42, 0.58] | 0.48 [0.41, 0.57] | 0.005 |

SD, standard deviation; IQR, interquartile range; MET, metabolic equivalent of task; HbA1c, hemoglobin A1C; mRNFL, macular retinal nerve fiber layer; HDL, high-density lipoprotein; VLDL, very low-density lipoprotein.

**Table 4.** Demographic and clinical characteristics in 2006–2010 of UK Biobank participants among the bottom 50% of glaucoma polygenic risk score.

| Characteristic | No Glaucoma | Glaucoma | p-value |
|---|---|---|---|
| Sample size (%) | 57,578 (98.7) | 780 (1.3) | |
| Sex - Male (%) | 26,887 (46.7) | 409 (52.4) | 0.002 |
| Age at recruitment, years, (mean (SD)) | 57 (8.0) | 60 (7.0) | <0.001 |
| Ethnicity (%) | | | <0.001 |
| White | 55,034 (96) | 690 (89) | |
| Asian | 1,129 (2.0) | 41 (5.3) | |
| Black | 574 (1.0) | 27 (3.5) | |
| Other | 841 (1.5) | 22 (2.8) | |
| Smoking Status (%) | | | 0.67 |
| Never | 200 (0.3) | 4 (0.5) | |
| Prefer not to answer | 31,593 (54.9) | 418 (53.6) | |
| Previous | 20,276 (35.2) | 287 (36.8) | |
| Current | 5,509 (9.6) | 71 (9.1) | |
| MET, minutes per week (mean (SD)) | 2,483 (2,450) | 2,612 (2,562) | 0.14 |
| Body mass index kg/m$^2$ (mean (SD)) | 27.4 (4.8) | 27.8 (4.7) | 0.021 |
| HbA1c, mmol/mol (mean (SD)) | 35.9 (5.7) | 38.1 (7.8) | <0.001 |
| Spherical Equivalent, diopter (mean (SD)) | –0.03 (2.1) | 0.01 (1.9) | 0.63 |
| Intraocular pressure, mmHg (mean (SD)) | 15.4 (2.4) | 17.1 (3.9) | <0.001 |
| mRNFL thickness, μm (mean (SD)) | 28.7 (1.7) | 28.6 (1.3) | 0.02 |
| Beta blocker use (%) | 4,322 (7.5) | 71 (9.1) | 0.11 |
| Caffeine intake, mg/day (mean (SD)) | 1,66.3 (67.4) | 166.6 (61.5) | 0.90 |
| Alcohol intake, g/week (median [IQR]) | 84 [41.7, 149.3] | 84 [42.5, 145.9] | 0.91 |
| Diabetes (%) | 3,209 (5.6) | 104 (13.3) | <0.001 |
| Oral steroid use (%) | 1,538 (2.7) | 39 (5.0) | <0.001 |
| Coronary Artery Disease (%) | 2,656 (4.6) | 64 (8.2) | <0.001 |
| Total cholesterol, mmol/l (median [IQR]) | 4.59 [3.98, 5.22] | 4.49 [3.76, 5.13] | 0.001 |
| Lactate, mmol/l (median [IQR]) | 3.95 [3.24, 4.75] | 3.84 [3.21, 4.69] | 0.16 |
| Concentration of Small HDL Particles, mmol/l (median [IQR]) | 0.0097 [0.0089, 0.011] | 0.0095 [0.0088, 0.010] | 0.002 |
| Cholesteryl esters in small HDL, mmol/l (median [IQR]) | 0.33 [0.30, 0.36] | 0.32 [0.30, 0.36] | 0.001 |
| Albumin, mmol/l (median [IQR]) | 39.4 [37.3, 41.45] | 38.9 [36.7, 40.8] | <0.001 |
| Total lipids in small HDL, mmol/l (median [IQR]) | 1.16 [1.06, 1.26] | 1.15 [1.05, 1.25] | 0.018 |
| Citrate, mmol/l (median [IQR]) | 0.065 [0.057, 0.074] | 0.065 [0.057, 0.073] | 0.52 |
| Pyruvate, mmol/l (median [IQR]) | 0.080 [0.06, 0.10] | 0.079 [0.061, 0.098] | 0.14 |
| Alanine, mmol/l (median [IQR]) | 0.29 [0.24, 0.35] | 0.29 [0.24, 0.35] | 0.95 |
| Phospholipids in Small HDL, mmol/l (median [IQR]) | 0.66 [0.60, 0.72] | 0.66 [0.60, 0.71] | 0.033 |

SD, standard deviation; MET, metabolic equivalents; HbA1c, hemoglobin A1C; mRNFL, macula region retinal nerve fiber layer; HDL, high density lipoprotein; VLDL, very low density lipoprotein.

**Table 5.** Metabolites associated with glaucoma among UK Biobank participants in the top decile and the bottom half of glaucoma polygenic risk score.

**Top 10% of glaucoma polygenic risk scorpe**

| Metabolites (Probit score) | Glaucoma | No Glaucoma | Adjusted p-value (NEF) |
|---|---|---|---|
| Lactate | –0.146 | 0.0239 | 8.8E-12 |
| Pyruvate | –0.137 | 0.0117 | 2.9E-10 |
| Citrate | –0.0693 | 0.0079 | 0.018 |
| Triglycerides in Very Large VLDL | 0.0606 | –0.0043 | 0.10 |
| Triglycerides in Chylomicrons and Extremely Large VLDL | 0.0572 | –0.0076 | 0.11 |
| Acetoacetate | 0.0744 | 0.0011 | 0.11 |
| Cholesteryl Esters in Small HDL | –0.0239 | 0.0376 | 0.12 |
| Cholesteryl Esters in Medium HDL | –0.0261 | 0.0276 | 0.13 |
| Alanine | –0.0612 | –0.0006 | 0.14 |
| Triglycerides in Large VLDL | 0.0629 | 0.0005 | 0.15 |

Potential confounders adjusted by regression include time since the last meal/drink (hours), age (years), age-squared (years-squared), sex, ethnicity (Asian, Black, White, and other), season, time of day of specimen collection (morning, afternoon, night), smoking status (never, past, and current smoker), alcohol intake, caffeine intake, physical activity (metabolic equivalent of task [MET] hours/week), body mass index (kg/m2), average systolic blood pressure (mm Hg), history of diabetes (yes or no), HbA1c (mmol/mol), history of coronary artery disease, systemic beta- blocker use, oral steroid use, and spherical equivalent refractive error (diopters).

of PRS quartile, with further stratification by holistic MRS in each PRS bin (*Figure 6*). The OR for PRS Q1 with MRS Q4 relative to PRS Q1 with MRS Q1 was modest (OR = 1.66; 95% CI = 1.15–2.42; p=0.0086). However, those with PRS Q4 (highest genetic risk) and MRS Q1 (lowest metabolic risk) had an OR of 11.7 (95% CI = 8.72–16.0; p<0.001) compared to PRS Q1 MRS Q1. This illustrates the importance of genetics in impacting glaucoma risk. Strikingly, the glaucoma OR further increased to 25.1 (95% CI = 18.8–34.1; p<0.001) for those with PRS Q4, MRS Q4 (combination of highest genetic and highest metabolic risk; $P_{interaction}$ = 0.019). Altogether, this demonstrates the strong prognostic utility of combining both PRS and MRS measurements for assessing glaucoma risk, particularly in individuals with a high genetic predisposition.

### Pyruvate supplementation lessens intraocular pressure and glaucoma

To functionally test the association between higher levels of pyruvate and resilience to glaucoma, we experimentally tested whether treatment with pyruvate induces resilience to IOP elevation and glaucoma in a human-relevant mouse model. Common variants in *LMX1B* are associated with IOP variation and the most common form of human glaucoma, primary open-angle glaucoma (POAG) (*Choquet et al., 2018*; *Gao et al., 2018*; *Gharahkhani et al., 2021*; *Khawaja et al., 2018*; *MacGregor et al., 2018*; *Shiga et al., 2018*). Rare Mendelian variants in *LMX1B* can produce early-onset ocular hypertension and open-angle glaucoma (OAG) (*Cross et al., 2014*). We have previously demonstrated that mice with a dominant mutation in *Lmx1b* (*Lmx1b$^{V265D/+}$*) develop IOP elevation and glaucoma (*Cross et al., 2014*; *Tolman et al., 2021*). Depending on the genetic background, this *Lmx1b$^{V265D}$* mutation induces either early-onset or later glaucoma (*Tolman et al., 2021*). Mutant mice with a C57BL/6 J strain background develop severe, early-onset IOP elevation and glaucoma. As pyruvate and its metabolites were associated with no glaucoma despite strong genetic predisposition in the UKBB cohort (highest decile of PRS), we tested the ability of dietary pyruvate to induce resilience against the *Lmx1b$^{V265D}$*-induced glaucoma on this C57BL/6 J genetic background. Pyruvate supplementation through drinking water substantially protected mice from IOP elevation and glaucoma. Pyruvate significantly protected against both anterior chamber deepening (ACD), a consequence of IOP elevation in mouse eyes (*Figure 7*), and IOP elevation itself (*Figure 7*). Importantly, pyruvate treatment protected against glaucomatous optic nerve degeneration (*Figure 7*). Together, our findings strongly support the role of endogenous pyruvate in conferring resilience against glaucoma even

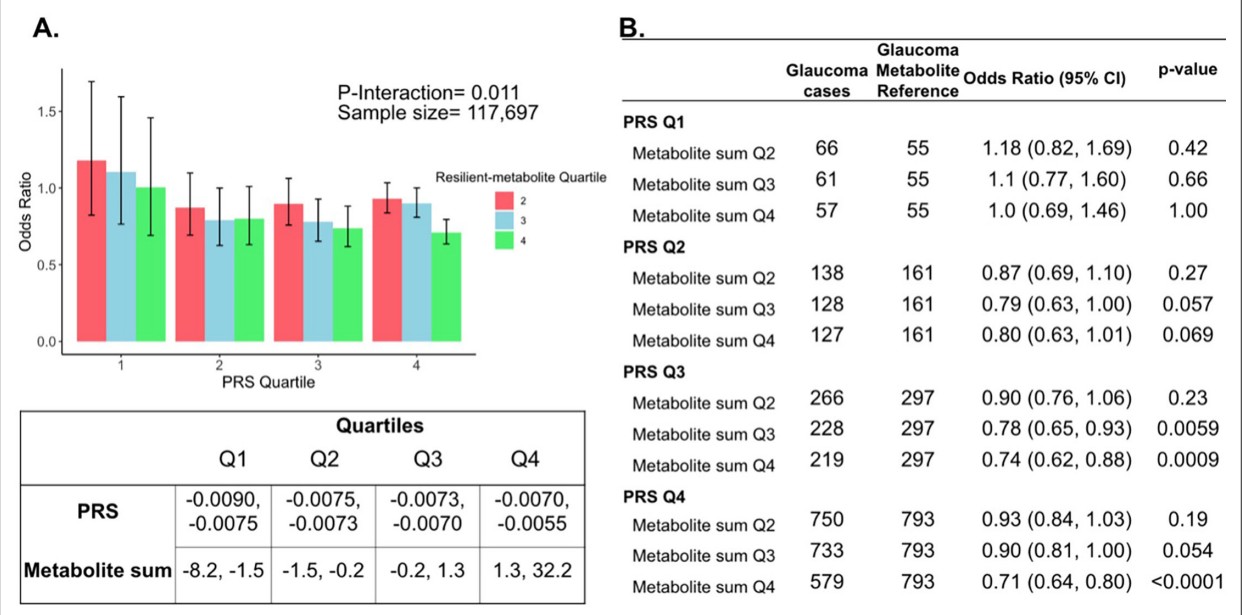

**A.**

P-Interaction= 0.011
Sample size= 117,697

Resilient-metabolite Quartile
- 2 (red)
- 3 (blue)
- 4 (green)

| Quartiles | | | | |
|---|---|---|---|---|
| | Q1 | Q2 | Q3 | Q4 |
| **PRS** | -0.0090, -0.0075 | -0.0075, -0.0073 | -0.0073, -0.0070 | -0.0070, -0.0055 |
| **Metabolite sum** | -8.2, -1.5 | -1.5, -0.2 | -0.2, 1.3 | 1.3, 32.2 |

**B.**

| | Glaucoma cases | Glaucoma Metabolite Reference | Odds Ratio (95% CI) | p-value |
|---|---|---|---|---|
| **PRS Q1** | | | | |
| Metabolite sum Q2 | 66 | 55 | 1.18 (0.82, 1.69) | 0.42 |
| Metabolite sum Q3 | 61 | 55 | 1.1 (0.77, 1.60) | 0.66 |
| Metabolite sum Q4 | 57 | 55 | 1.0 (0.69, 1.46) | 1.00 |
| **PRS Q2** | | | | |
| Metabolite sum Q2 | 138 | 161 | 0.87 (0.69, 1.10) | 0.27 |
| Metabolite sum Q3 | 128 | 161 | 0.79 (0.63, 1.00) | 0.057 |
| Metabolite sum Q4 | 127 | 161 | 0.80 (0.63, 1.01) | 0.069 |
| **PRS Q3** | | | | |
| Metabolite sum Q2 | 266 | 297 | 0.90 (0.76, 1.06) | 0.23 |
| Metabolite sum Q3 | 228 | 297 | 0.78 (0.65, 0.93) | 0.0059 |
| Metabolite sum Q4 | 219 | 297 | 0.74 (0.62, 0.88) | 0.0009 |
| **PRS Q4** | | | | |
| Metabolite sum Q2 | 750 | 793 | 0.93 (0.84, 1.03) | 0.19 |
| Metabolite sum Q3 | 733 | 793 | 0.90 (0.81, 1.00) | 0.054 |
| Metabolite sum Q4 | 579 | 793 | 0.71 (0.64, 0.80) | <0.0001 |

**Figure 5.** Interaction of three putative resilient metabolites (lactate, pyruvate, and citrate) and polygenic risk score (PRS) on glaucoma risk. (**A**) The bar chart shows the interaction of resilient probit-transformed metabolite sum with glaucoma genetic predisposition in each PRS quartile. In each glaucoma PRS quartile, the lowest metabolic sum quartile (Q1) is the metabolite reference group used to calculate the odds ratios. Each color represents resilient-metabolite sum quartiles (red = second quartile; blue = third quartile; and green = fourth quartile). Error bars show 95% confidence interval (CI). The table under the bar chart shows the ranges for the PRS and metabolite sum value quartiles. (**B**) The table shows odds ratios for glaucoma by PRS, and putative resilient metabolite sum within various quartiles. The number of glaucoma cases within each resilient metabolite sum quartile and the number of glaucoma cases in the first quartile of resilient metabolite sum (Q1, labeled as glaucoma metabolite reference) are used to calculate the odd ratios. This analysis is adjusted for time since the last meal/drink (hours), age (years), age-squared (years-squared), sex, ethnicity (Asian, Black, White, and other), season, time of day of specimen collection (morning, afternoon, night), smoking status (never, past, and current smoker), alcohol intake, caffeine intake, physical activity (metabolic equivalent of task [MET] hours/week), body mass index (kg/m²), average systolic blood pressure (mm Hg), history of diabetes (yes or no), HbA1c (mmol/mol), history of coronary artery disease, systemic beta-blocker use, oral steroid use, and spherical equivalent refractive error (diopters).

countering strong genetic predisposition. Furthermore, they show that pyruvate can act as a potent resilience factor against IOP elevation and glaucoma when delivered orally.

## Discussion

In this large cohort study, we found that the inclusion of metabolite data from an NMR platform only marginally improved various glaucoma prediction algorithms (*Figure 3*). Nonetheless, using an agnostic approach, we found that lactate, pyruvate, and citrate levels collectively were associated with a 29% reduced risk of glaucoma among those in the top quartile of glaucoma genetic predisposition (*Figure 5*). We showed that pyruvate supplementation reduced glaucoma incidence in a human-relevant genetic mouse model (*Figure 7*). Furthermore, the interaction between glaucoma PRS and an MRS based on 168 metabolites provided synergistic predictive information regarding glaucoma risk (*Figure 6*). The statistical interaction we demonstrate between metabolite scores and PRS-derived glaucoma risk helps identify individuals who, despite high genetic risk, are less likely to develop glaucoma due to favorable metabolic profiles or, on the contrary, are more likely to develop glaucoma and should be monitored at an earlier age. Overall, these findings indicate that an MRS may provide clinically useful measures of metabolic status and improve patient risk stratification in glaucoma.

While earlier studies (*Mabuchi et al., 2017*; *Tham et al., 2015*; *Zanon-Moreno et al., 2017*) showed modest discriminatory powers for a glaucoma PRS, recent larger genome-wide association studies reported significantly improved risk prediction (*Craig et al., 2020*; *Han et al., 2019*; *Gao et al., 2019*; *Nannini et al., 2018*). However, no studies have examined the utility of incorporating metabolite data into glaucoma prediction algorithms combined with a PRS. Future work is needed to understand how non-genetic factors, known to alter glaucoma risk or IOP levels, like air pollution,

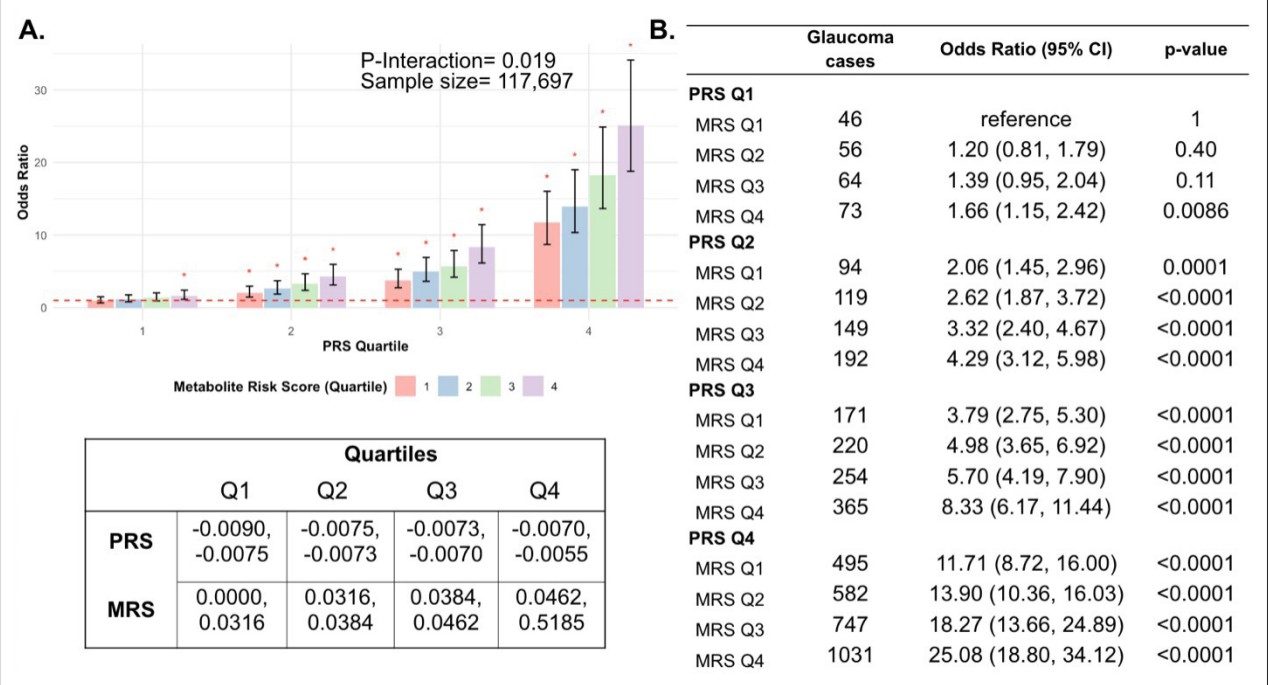

| | Glaucoma cases | Odds Ratio (95% CI) | p-value |
|---|---|---|---|
| **PRS Q1** | | | |
| MRS Q1 | 46 | reference | 1 |
| MRS Q2 | 56 | 1.20 (0.81, 1.79) | 0.40 |
| MRS Q3 | 64 | 1.39 (0.95, 2.04) | 0.11 |
| MRS Q4 | 73 | 1.66 (1.15, 2.42) | 0.0086 |
| **PRS Q2** | | | |
| MRS Q1 | 94 | 2.06 (1.45, 2.96) | 0.0001 |
| MRS Q2 | 119 | 2.62 (1.87, 3.72) | <0.0001 |
| MRS Q3 | 149 | 3.32 (2.40, 4.67) | <0.0001 |
| MRS Q4 | 192 | 4.29 (3.12, 5.98) | <0.0001 |
| **PRS Q3** | | | |
| MRS Q1 | 171 | 3.79 (2.75, 5.30) | <0.0001 |
| MRS Q2 | 220 | 4.98 (3.65, 6.92) | <0.0001 |
| MRS Q3 | 254 | 5.70 (4.19, 7.90) | <0.0001 |
| MRS Q4 | 365 | 8.33 (6.17, 11.44) | <0.0001 |
| **PRS Q4** | | | |
| MRS Q1 | 495 | 11.71 (8.72, 16.00) | <0.0001 |
| MRS Q2 | 582 | 13.90 (10.36, 16.03) | <0.0001 |
| MRS Q3 | 747 | 18.27 (13.66, 24.89) | <0.0001 |
| MRS Q4 | 1031 | 25.08 (18.80, 34.12) | <0.0001 |

**Quartiles**

| | Q1 | Q2 | Q3 | Q4 |
|---|---|---|---|---|
| **PRS** | -0.0090, -0.0075 | -0.0075, -0.0073 | -0.0073, -0.0070 | -0.0070, -0.0055 |
| **MRS** | 0.0000, 0.0316 | 0.0316, 0.0384 | 0.0384, 0.0462 | 0.0462, 0.5185 |

P-Interaction= 0.019
Sample size= 117,697

**Figure 6.** Interaction of the holistic metabolite risk score (n=168 metabolites) and polygenic risk score (PRS) on glaucoma risk. (**A**) The bar chart plots the odds ratio of glaucoma as a function of holistic probit-transformed MRS quartile with further stratification by glaucoma PRS in each MRS bin. The lowest quartile of glaucoma PRS and MRS is the reference group (see dotted red line) for the entire population. Each color represents the MRS quartiles (red = first quartile; blue = second quartile; green = third quartile; and purple = fourth quartile). Error bars show the 95% confidence interval (CI). The table under the bar chart shows the ranges for the PRS and MRS quartiles. (**B**) Table showing odds ratios for glaucoma by polygenic risk score (PRS) and MRS within various quartiles. The number of glaucoma cases within each MRS and the number of glaucoma cases in PRS Q1 and MRS Q1 are used to calculate the odds ratios. This analysis is adjusted for time since the last meal/drink (hours), age (years), age-squared (years-squared), sex, ethnicity (Asian, Black, White, and other), season, time of day of specimen collection (morning, afternoon, night), smoking status (never, past, and current smoker), alcohol intake, caffeine intake, physical activity (metabolic equivalent of task [MET] hours/week), body mass index (kg/m²), average systolic blood pressure (mm Hg), history of diabetes (yes or no), HbA1c (mmol/mol), history of coronary artery disease, systemic beta-blocker use, oral steroid use, and spherical equivalent refractive error (diopters).

psychological stress, physical activity, and dietary factors (*Stuart et al., 2023b*) could change the MRS and alter high glaucoma genetic predisposition.

Previous studies have explored metabolomics to identify biomarkers for OAG. A systematic review analyzing 13 studies identified a total of 144 metabolites, of which 12 were associated with OAG (*Wang et al., 2021*). Among studies using plasma samples, four metabolic pathways were significantly enriched: sphingolipid metabolism, arginine and proline metabolism, and beta-alanine metabolism. In another study across three US cohorts and the UKBB, higher levels of plasma diglycerides and triglycerides were adversely associated with glaucoma (*Zeleznik et al., 2023*) This is consistent with our observation that lower triglyceride levels were associated with glaucoma resilience, albeit with less significance compared to the three glycolysis-related metabolites. The biological mechanism is suggested to be the association of hypertriglyceridemia with increased blood viscosity and elevated IOP (*Madjedi et al., 2022*; *Wang and Bao, 2019*; *Pertl et al., 2017*; *Rasoulinejad et al., 2015*). Consistent with this, a meta-analysis reported that individuals with glaucoma had a mean triglyceride level 14.2 mg/dl higher than those without the disease (*Pertl et al., 2017*).

While studies have identified biomarkers associated with glaucoma, to the best of our knowledge, there has been no investigation on biomarkers that are associated with being resilient to a high glaucoma genetic predisposition. We identified elevated levels of lactate, pyruvate, and citrate among individuals without glaucoma despite high genetic risk. The retina commonly utilizes energy produced from glycolysis and oxidative phosphorylation. Therefore, glycolysis end products (pyruvate and lactate) are important energy sources for retinal ganglion cells (RGCs) (*Williams and Casson, 2024*) RGCs express monocarboxylate transporters, which facilitate the uptake of extracellular pyruvate and

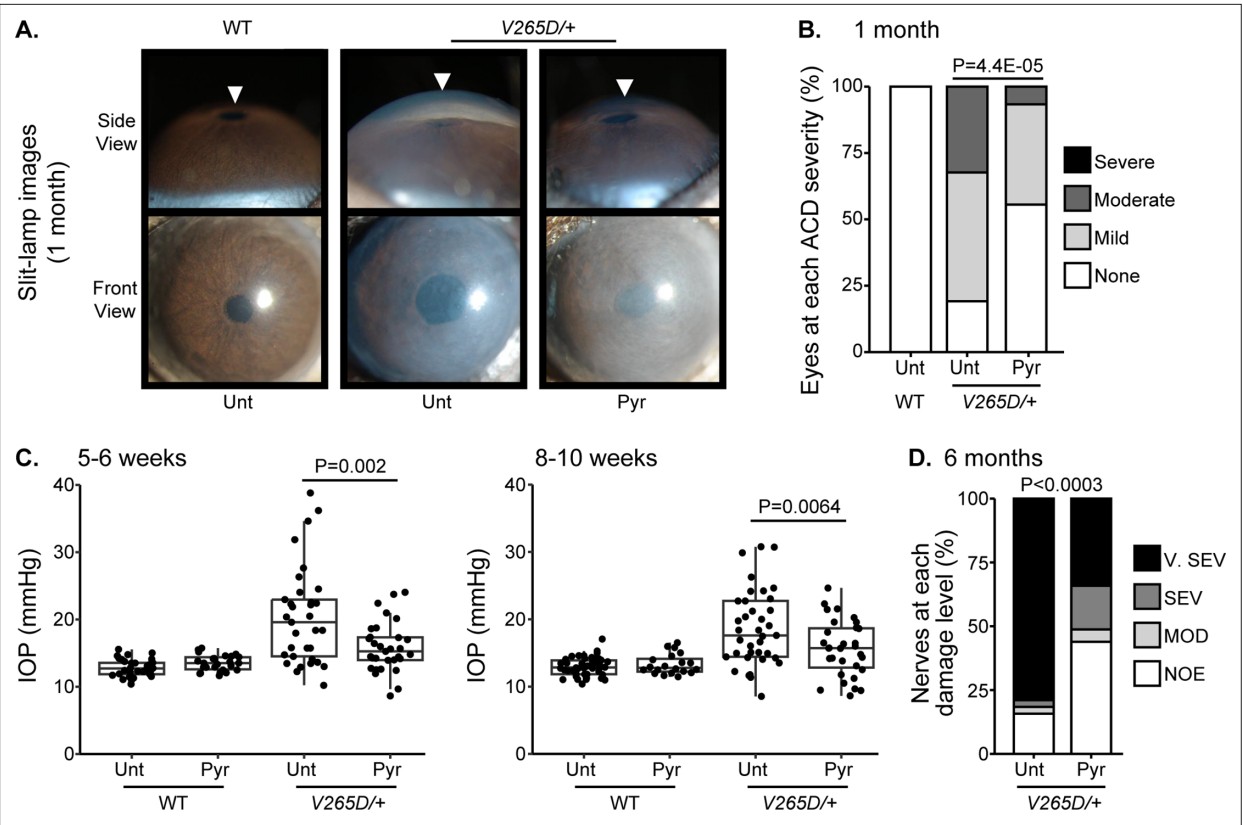

**Figure 7.** Pyruvate treatment protects from intraocular pressure (IOP) elevation and glaucoma. (**A**) Representative photos of eyes from mice of the indicated genotypes and treatments (Unt = untreated, Pyr = pyruvate treated). Lmx1b is expressed in the iris and cornea, so Lmx1b[V265D] mutant eyes have primary abnormalities of the iris and cornea. This includes corneal haze, which is present before IOP elevation in many eyes and likely reflects a direct transcriptional role of LMX1B in collagen gene expression. Lmx1b[V265D] mutant eyes also develop anterior chamber deepening (ACD), a sensitive indicator of IOP elevation in mice. The WT and pyruvate-treated mutant eyes have shallow anterior chambers, while the untreated mutant eye has a deepened chamber (arrowheads). (**B**) Distributions of ACD are based on a previously defined scoring system.[31] Groups are compared by Fisher's exact test. n > 40 eyes were examined in each group. (**C**) Boxplots of IOP (interquartile range and median line) in WT and mutant eyes. Pyruvate treatment significantly lessens IOP elevation in mutants compared to untreated mutant controls. Groups were compared by ANOVA followed by Tukey's honestly significant difference. n > 30 eyes were examined in each Lmx1b[V265D] mutant group, and n > 20 eyes were examined in WT groups. (**D**) Distributions of damage based on analysis of para-phenylenediamine (PPD)-stained optic nerve cross sections from 6-mo-old mice (Methods). Pyruvate treatment lessened the incidence of glaucoma (Fisher's exact test). No glaucoma was found in WT mice. Geno = genotype. n = 38–41 nerves examined per group. NOE = no glaucoma, MOD = moderate. SEV = severe, and V. SEV = very severe (see Methods).

lactate, improving energy homeostasis, neuronal metabolism, and survival (*Harun-Or-Rashid et al., 2020*; *Pappenhagen et al., 2019*) Lactate is critical for RGC survival during periods of glucose deprivation *Vohra et al., 2019a*; *Rajala and Rajala, 2023* and bioenergetic insufficiency contributes to RGC loss in glaucoma (*Williams et al., 2017b*; *Inman and Harun-Or-Rashid, 2017*; *Williams et al., 2017a*; *Chen et al., 2013*). Finally, a study found that baseline lactate levels were significantly decreased in patients with normal-tension glaucoma compared to non-glaucomatous controls (*Vohra et al., 2019b*).

Pyruvate plays a central role in energy metabolism by linking glycolysis to the TCA cycle, ultimately driving ATP production by the electron transport chain. Pyruvate is converted into acetyl-CoA, which fuels the TCA cycle, and into lactate, a key molecule that regulates metabolism and serves as an important energy source (*Brooks, 2020*). The TCA cycle is tightly regulated to balance biosynthesis (producing metabolic precursors for lipids, carbohydrates, amino acids, nucleic acids, and co-factors) and energy production, adjusting dynamically to cellular needs (*Arnold and Finley, 2023*). Beyond its metabolic functions, pyruvate also acts as an antioxidant, scavenging reactive oxygen species (ROS), inducing the expression of the antioxidant response control gene *Nrf2*, and promoting beneficial autophagy (*Jiménez-Moreno et al., 2023*; *Kim et al., 2013*; *Shin et al., 2012*). Its neuroprotective effects on RGCs have been demonstrated both in cell culture and in an induced model of glaucoma

in rats (*Harder et al., 2020*). Additionally, pyruvate combined with nicotinamide has been shown to enhance optic nerve protection in mice (*Guymer et al., 2018*). Importantly, the current study extends this evidence by demonstrating that pyruvate alone provides strong protection against glaucoma in a mouse model carrying a mutation in the ortholog of a human POAG gene. Furthermore, it extends pyruvate's protective effects to the ocular drainage tissues, mitigating IOP elevation. Notably, although our validation data demonstrates the neuroprotective effects of exogenous pyruvate, it remains possible that endogenously produced pyruvate within ocular tissues may also contribute to RGC protection. A randomized phase II clinical trial that tested pyruvate and nicotinamide in POAG patients found improved visual function in glaucoma patients compared to the placebo group (*De Moraes et al., 2022*). Together, both preclinical and clinical evidence support the role of pyruvate as a protective factor against glaucoma.

Citrate is a key intermediate in the TCA cycle and is important for aerobic energy production. Elevated citrate levels might reflect enhanced energy supply and antioxidative capacity of ocular tissues, potentially protecting RGCs against oxidative stress (*Williams et al., 2017a*; *Wu et al., 2019*; *Morganti et al., 2020*; *Catalina-Rodriguez et al., 2012*; *Ohanele et al., 2024*). Decreased levels of citrate may indicate increased usage or mitochondrial dysfunction (*Chhimpa et al., 2023*) Studies have found lower plasma citrate levels in glaucoma patients among adults and children (*Fraenkl et al., 2011*; *Michalczuk et al., 2017*). A study measuring plasma citrate with a cutoff of 110 μmol/L to detect glaucoma had a sensitivity of 66.7% and a specificity of 71.4% (*Fraenkl et al., 2011*). These findings corroborate the protective effects of citrate in conferring resilience to glaucoma. Similar to pyruvate, plasma citrate may also play a role in retinal homeostasis, highlighting the need to disentangle serum derived from intraocularly synthesized metabolites.

We also identified several lipoproteins associated with resilience, most notably cholesteryl esters in small high-density lipoprotein (HDL) and medium HDL. Although these lipoproteins did not reach an adjusted *P*-value <0.05 and thus remain suggestive biomarkers rather than definitive, this is consistent with studies describing the neuroprotective effect of HDL in the glaucoma pathophysiology (*Nusinovici et al., 2022*; *Masson et al., 2023*). Cholesteryl esters and triglycerides make up the core of lipoproteins, which transport lipids in the blood. Lipoproteins are characterized by their size, where HDL ranges from 8 to 12 nm in diameter, and low-density lipoprotein (LDL) ranges from 18 to 25 nm in diameter. The retinal pigment epithelium can take up lipoproteins *Increased High-Density Lipoprotein Levels Associated with Age-Related Macular Degeneration: Evidence from the EYE-RISK and European Eye Epidemiology Consortia - ScienceDirect, 2024*; *Park and Park, 2020* and dysregulation of lipid metabolism can result in the accumulation of lipids and oxidative stress in the retina, resulting in RNFL thinning (*Zeleznik et al., 2023*; *Pasquale et al., 2023*; *Ana et al., 2023*; *Fu et al., 2019*; *Yuksel et al., 2023*). Thus, HDL has antioxidant properties to protect RGCs by facilitating cholesterol efflux from the retina (*Kelly et al., 2020*; *Morvaridzadeh et al., 2024*). Furthermore, as HDL analytes can cross blood-ocular barriers (*Hamel et al., 2024*) there is a plausible route for serum-derived HDL to influence RGC homeostasis. Nonetheless, the relative contributions of circulating lipoproteins versus local synthesis within ocular tissues remain unclear and warrant further investigation. Notably, genes implicated by large cross-ancestry and European subset GWAS meta-analyses of POAG risk are enriched in processes related to apolipoprotein binding and lipid transport (*Gharahkhani et al., 2021*; *Hamel et al., 2024*). Incorporating ocular physiology and blood-retinal barrier considerations when interpreting lipoproteins as potential resilience biomarkers will be critical for future studies aimed at understanding and therapeutically targeting increased glaucoma risk.

This study had several strengths. First, the UKBB is a large cohort study with detailed covariate information on demographics, clinical characteristics, and glaucoma endophenotypes, allowing for better glaucoma risk profiling. Second, we used a PRS calculated from the large (~600,000 individuals) multi-trait analysis of genome-wide association studies on glaucoma and endophenotypes, including optic nerve head structural features and IOP data (*Craig et al., 2020*). Finally, we supported our biobank analyses with functional tests in a disease-relevant mouse model, identifying a protective effect of pyruvate supplementation in animals susceptible to glaucoma. The data showing that pyruvate protected against IOP elevation (a key causative risk factor for glaucoma) and glaucomatous nerve damage in a mouse model, as well as pyruvate's biochemical relatedness to lactate and citrate regarding energy metabolism, mitigate against concerns about false discovery, uncontrolled confounding, reverse causation, and collinearity among biomarkers.

A major limitation of our study is that it includes a predominately European population, with an underrepresentation of African, Asian, and mixed American genetic ancestral groups. Nonetheless, we chose to analyze Europeans and non-Europeans together to make the results as generalizable as possible. A second limitation is that glaucoma ascertainment in the UK Biobank is based on self-reported diagnoses and hospital records rather than comprehensive ophthalmologic examinations. Nonetheless, it is reassuring that the prevalence of glaucoma in our sample (~4%) is similar to a directly performed disease burden estimate in a comparable, albeit slightly older, United Kingdom sample (2.7%) (*Minassian et al., 2000*). In addition, a subset of UKBB participants have received glaucoma treatment, which could alter metabolite levels and could lead to an underestimate of the protective effects of metabolites. Moreover, the UKBB study includes a limited set of 168 metabolites, which is not inclusive of the comprehensive set of metabolites, known and unknown. This study also only focuses on plasma metabolites and does not account for ocular metabolite profiles. Due to the cross-sectional design of our study, establishing temporality is challenging. As a result, it remains unclear whether the observed metabolic changes contribute to the onset of glaucoma or arise as a consequence of the disease. However, the mouse data support a causal role for pyruvate as a factor that mitigates high genetic risk. This study's exploratory approach– investigating metabolite associations in an agnostic, hypothesis-free manner– underscores the need for replication. Regarding the mouse studies, we performed ad-lib pyruvate dosing and examined only a single human-related mutation, whereas individuals with a genetic predisposition to glaucoma typically carry a high cumulative burden of multiple common glaucoma-associated variants. Lastly, neither the mouse studies, nor the human data accurately inform the exact dosing of TCA cycle biomarkers needed to mitigate glaucoma risk. We find considerable overlap between TCA cycle biomarkers for those with and without glaucoma in the top decile of glaucoma risk (*Appendix 2—table 2* and *Appendix 2—figure 2*).

Overall, we have shown that plasma metabolites marginally enhance the predictive power of a glaucoma PRS and may have a role in identifying those who may be resilient to glaucoma despite high genetic predisposition. Our study identified protective metabolites linked to glycolysis and mitochondrial function, suggesting important pathophysiology in glaucoma. One of these metabolites of significance is pyruvate. Oral pyruvate supplementation substantially protected against IOP elevation and conferred resilience against glaucoma. Thus, these results open new avenues for therapeutic strategies of pyruvate and its metabolites as resilience factors against glaucoma in genetically predisposed individuals.

## Methods

### Study design

Our initial human analysis was conducted in three steps. First, we assessed whether plasma metabolite data alone can predict glaucoma risk with receiver operating characteristic (ROC) curves in the UK Biobank. We then integrated the metabolite data into a polygenic risk score (PRS)-based glaucoma risk assessment model to see if metabolites could enhance glaucoma prediction. Second, we focused on identifying a metabolomic signature of resilience to high glaucoma PRS by comparing plasma metabolites in glaucoma cases versus participants without glaucoma (henceforth referred to as resilient participants) from the top 10% of the glaucoma PRS. Third, we explored the interactions between metabolic scores and PRS in modifying glaucoma risk.

### Study population and data collection

The UKBB is a prospective cohort study of over half a million participants aged 37–73 years old at recruitment across the United Kingdom from 2006 to 2010. Participants were recruited through the National Health Service registers from 22 assessment centers, where they signed electronic informed consent to participate. Participants then completed in-depth touchscreen questionnaires and trained staff-led interviews, performed body measurements, and provided biological samples (*Sudlow et al., 2015*). Additional outcomes are available from data linkage to hospital episode statistics, the death register, and primary data (*Sudlow et al., 2015*; *Bycroft et al., 2018*; *Zeleznik et al., 2022*). Biological samples collected from these participants, including blood, urine, and saliva specimens, were used to generate genetic, metabolomic, and proteomic data (*Elliott et al., 2008*). Details of the UKBB study design and population are described online (https://www.ukbiobank.ac.uk). The UKBB study received

approval from the National Health Service North West Multicentre Research Ethics Committee (reference number 06/MRE08/65) and the National Information Governance Board for Health and Social Care. This research, conducted under UK Biobank application number 36741, adhered to the tenets of the Declaration of Helsinki.

The UKBB included 502,613 participants, of which 173,679 participants had self-reported glaucoma data, ICD-coded, or previous glaucoma laser/surgical therapy (34.6%) at baseline. A subset consisting of 117,698 participants with metabolomic data and genetic profiling comprise the study population (*Figure 1*). The glaucoma cases were defined based on those with self-reported glaucoma, in which participants selected 'glaucoma' when they completed a touchscreen questionnaire with the question, 'Has a doctor told you that you have any of the following problems with your eyes?'. Glaucoma cases also included participants who reported a history of glaucoma surgery or laser therapy on the questionnaire or if an International Classification of Diseases (ICD) code for glaucoma (ICD nineth revision: 365.* [excluding 365.0]; ICD 10th revision: H40.* [excluding H40.0] and H42.*) was carried in linked Hospital Episode Statistics before the baseline assessment. The described approach to identifying glaucoma has been strongly supported by prior research publications (*Khawaja et al., 2018*; *MacGregor et al., 2018*).

## Metabolite profiling

In the UKBB, a high-throughput nuclear magnetic resonance (NMR)-based biomarker platform (Nightingale Health Ltd; Helsinki, Finland) was used to measure the metabolomic profile in the randomly selected non-fasting EDTA plasma samples from a subset of participant. (*Julkunen et al., 2021*). In contrast to liquid chromatography/mass spectroscopy, NMR spectroscopy produces distinctive spectral shapes for molecules containing hydrogen atoms (*Takis et al., 2019*) The AUCs are proportional to the concentration of each molecule based on chemical shifts and J coupling split patterns derived from quantum mechanics (*Soininen et al., 2015*; *Mihaleva et al., 2014*). The NMR platform contains data on 249 metabolic biomarkers (168 absolute levels and 81 ratio measures), including a subset of 36 biomarkers (27 absolute levels and 9 ratio measures) certified for broad diagnostic use by the European Union (EU) (*Julkunen et al., 2021*). In this study, we focused on the 168-metabolite and the 27-metabolite sets for analysis (for a complete list of quantified metabolites, refer to Appendix 1, Supplementary Methods). Individual metabolite values were transformed to probit scores to standardize their range and reduce the impact of skewed distributions.

## Construction of PRS for glaucoma

The glaucoma PRS used in this study was from a multi-trait analysis of GWAS (MTAG) on glaucoma (*Craig et al., 2020*) This PRS was derived based on GWAS data from glaucoma (7,947 cases and 119,318 controls) and its endophenotypes: optic nerve head structure using vertical cup-disc ratio (VCDR) (including additional data from 67,040 UKBB and 23,899 International Glaucoma Genetics Consortium (IGGC) participants), and intraocular pressure (IOP, including additional data on 103,914 UKBB and 29,578 IGGC participants). The glaucoma PRS was constructed using single nucleotide polymorphisms (SNPs) with MTAG p-values ≤0.001, resulting in 2,673 uncorrelated SNPs after LD-clumping at $r^2$=0.1, and was found to have a predictive ability with an AUC of 0.68 (95% confidence interval, CI, [0.67–0.70]) and 0.80 accounting for age, sex, family history, and PRS tested in the Australian and New Zealand Registry of Advanced Glaucoma cohort.

## Model building/covariates

We first assessed whether the inclusion of probit-transformed metabolite data enhances glaucoma prediction algorithms. We adjusted for factors involving major determinants of variability in metabolites, POAG-established and suspected risk factors, and other comorbidities. In model 1, we considered metabolites only. Model 2 incorporated additional demographic covariates, including age (years), sex, genetic ancestry, season, time of day of specimen collection (morning, afternoon, night), and fasting time (hours). Model 3 incorporated covariates in model 2 and additional variables accounting for comorbidities potentially related to glaucoma including smoking status (never, past, and current smoker), alcohol intake (g/week) (*Stuart et al., 2023a*), caffeine intake (mg/day) (*Kim et al., 2021*), physical activity (metabolic equivalent of task [MET], hours/week), body mass index (kg/m$^2$), average systolic blood pressure (mm Hg), history of diabetes, hemoglobin A1C (HbA1c, mmol/

mol), self-reported history of coronary artery disease, systemic beta-blocker use, oral steroid use, and mean spherical equivalent refractive error (diopters) across both eyes. Model 4 incorporated covariates in model 3 and a glaucoma PRS. We compared these models with and without the metabolites to determine the predictive ability of the metabolites above and beyond a glaucoma PRS and known risk factors. Missing values were imputed with medians for numeric variables and modes for factor variables.

## Statistical analysis

### Metabolite-based predictive modeling

Multivariable logistic regression models with regularization (implemented with *glmnet*) were built to investigate the associations of metabolic biomarkers and the risk of glaucoma. We used four sequential models, adjusting for covariates as described above. For each model, we examined three groupings: no metabolites, the 27 metabolites corresponding to EU-stamped validated markers, and 168 metabolites corresponding to all the measured plasma metabolites captured in the UKBB. We used regularization to address collinearity, which reduced the set of metabolites and other covariates considered in each model (reported in *Figure 3—source data 1*). To examine model accuracy for predicting glaucoma prevalence, we utilized ROC curves and associated AUC measurements as a metric for model performance. We performed fivefold cross-validation and split data into 80% used for training and 20% used for evaluation. All metabolites' values were probit transformed and used as continuous variables (per 1 standard deviation (SD) increase). Additionally, we stratified glaucoma by age, ethnicity, and sex. The DeLong test was utilized to examine the statistical significance of AUC differences between models, with a threshold of $p < 0.05$ as statistically significant.

### Identifying resilience metabolites to high glaucoma polygenic risk scores

To identify individual metabolites associated with resilience to a high glaucoma PRS, we first performed a logistic regression model to obtain metabolite residuals from probit-transformed concentration data adjusting for the following variables: age, age-squared, time since the last meal/drink (≤4, 5–8, and 9+ hr), sex, ethnicity (Asian, Black, White, and mixed/other), season, time of day of specimen collection, smoking status (never, past, and current smoker), alcohol intake, caffeine intake, physical activity (metabolic equivalent of task [MET] hours/week), body mass index ($kg/m^2$), average systolic blood pressure (mm Hg), history of diabetes, HbA1c (mmol/mol), history of coronary artery disease, systemic beta-blocker use, oral steroid use, and spherical equivalent refractive error (diopters). Metabolites associated with glaucoma were nominated by performing a t-test comparing residuals in participants with and without glaucoma to identify metabolites significantly associated with glaucoma in both the top 10% and the bottom 50% of glaucoma PRS score. This stratification balanced case counts to enable comparison of groups with adequate sample size (780 glaucoma cases in the bottom 50% of PRS score and 1,693 glaucoma cases in the top 10% of PRS score). To account for covariance in metabolite abundance, we corrected for multiple comparisons using the number of effective tests (NEF) method (*Gao et al., 2008*). First, a correlation matrix was generated using pairwise complete observations to address any missing values. Eigenvalue decomposition was performed on the correlation matrix to capture the principal components. Principal components contributing more than 1% of the total variance were calculated, yielding 9 significant components, capturing 91.6% of the variance. Thus, we applied a Bonferroni adjustment based on the number of significant components (n=9) to calculate the adjusted p-values accounting for multiple hypotheses. p-values are considered statistically significant if the NEF-adjusted p-value was <0.05 (*Gao et al., 2008*). Given this was an exploratory analysis, NEF-adjust p-values <0.2 were considered worthy of additional analysis.

### Construction of metabolite risk scores

We constructed MRS, which were calculated as a weighted sum of each metabolic biomarker from coefficients in the 168-metabolite base model. Specifically, we utilized a logistic regression model (implemented with a *binomial glm*) using probit transformed metabolite values to predict the prevalence of glaucoma using fivefold cross-validation. The resulting beta values were extracted and are reported in *Appendix 2—table 1*. To determine if there is an interaction between glaucoma MRS and PRS, we utilized extracted model coefficients and corresponding significance from a binomial generalized linear model. We fit this model using the formula *Glaucoma ~PRS * MRS + PRS + MRS +*

*Covariates.* We classified participants into 16 groups by quartiles of MRS and PRS. Within each PRS quartile, we used MRS quartile 1 as the reference for comparison. Additionally, we used the group in both PRS quartile 1 and MRS quartile 1 as the overall reference group for all comparisons among the groups. A chi-squared test was used to assess statistical significance between odds ratios (ORs) with 95% confidence intervals (CIs).

To investigate whether age modifies the associations of resilience metabolites with glaucoma, we conducted a stratified analysis and assessed a three-way interaction term of age ($\geq$ or <58 years old, based on median age), the sum of resilience metabolite levels, and PRS using a Wald test for individuals in the top 10% PRS. All statistical analyses and plots were produced using R version 4.2.1 (R Foundation for Statistical Computing, Vienna, Austria). All statistical tests were two-sided.

## Animal husbandry and ethics statement

Experimental mice had a C57BL/6 J (Jackson Laboratory Stock #000664) genetic background. The $Lmx1b^{V265D}$ (alias $Lmx1b^{lcst}$; European Mouse Mutant Archive, EM:00114) mutation is previously reported to cause high IOP and glaucoma in mice, with IOP becoming elevated in some eyes during the first months after birth (*Cross et al., 2014*; *Tolman et al., 2021*; *Thaung et al., 2002*). The mutation was backcrossed to the C57BL/6 J strain for at least 30 generations. All mice were treated per the Association for Research in Vision and Ophthalmology's statement on the use of animals in ophthalmic research. The Institutional Animal Care and Use Committee of Columbia University approved all experimental protocols performed (protocol IDs: AC-AABE9554 and AC-AABD0557). Mice were maintained on PicoLab Rodent Diet 20 (5053, 4.5% fat) and provided with reverse osmosis-filtered water. Mutant and control littermates were housed together in cages containing ¼-inch corn cob bedding, covered with polyester filters. The animal facility was maintained at a constant temperature of 22 °C with a 14 hours light and 10 hours dark cycle.

## Genotyping mice

$Lmx1b^{V265D}$ and $Lmx1b^+$ genotypes were determined by direct Sanger sequencing of a specific PCR product. Genomic DNA was PCR amplified with forward primer 5′-CTTTGAGCCATCGGAGCTG-3′ and reverse primer 5′-ATCTCCGACCGCTTCCTGAT-3′ using the following program: (1) 94 °C for 3 minutes; (2) 94 °C for 30 seconds; (3) 57 °C for 30 seconds; (4) 72 °C for 1 minute; (5) repeat steps 2–4 for 35 times; and (6) 72 °C for 5 minutes. PCR products were purified and sequenced by the Genewiz (Azenta Life Sciences).

## Slit-lamp examination

Anterior segments were examined approximately every 2 weeks between 1 month and 3 months of age and examined at monthly intervals between 3 months of age and harvest age (6 months). Balanced groups of males and females were examined as previously published (*Cross et al., 2014*; *Tolman et al., 2021*). Photographs were taken with a 40x objective lens. Anterior chamber deepening, a sign of exposure to high IOP, was graded based on a semiquantitative scale of no deepening (not present), mild, moderate, or severe, as previously published (*Tolman et al., 2021*). Under examination by either a slit-lamp or dissection scope (during IOP procedure), anterior chamber deepening was only detected in *Lmx1b* mutant groups. Groups were compared statistically by Fisher's exact test. There were n>40 eyes (biological replicates) were examined in each group.

## Intraocular pressure measurement

IOP was measured with the microneedle method as previously described in detail (*John, 1997*; *Savinova et al., 2001*). Before cannulation, mice were acclimatized to the procedure room and anesthetized via an intraperitoneal injection of a mixture of ketamine (99 mg/kg; Ketlar, Parke-Davis, Paramus, NJ, USA) and xylazine (9 mg/kg; Rompun, Phoenix Pharmaceutical, St Joseph, MO, USA) immediately prior to IOP assessment, a procedure that does not alter IOP in the experimental window (*Savinova et al., 2001*). IOP was measured at both 5–6 weeks and 8–10 weeks of age in WT and $Lmx1b^{V265D/+}$ eyes. Balanced groups of males and females were examined. During each IOP measurement period, the eyes of independent WT B6 mice were assessed in parallel with experimental mice as a methodological control to ensure proper calibration and equipment function. Experimenters were masked to

mouse genotype and drug treatment. *Lmx1b* mutant groups were compared by ANOVA followed by Tukey's honestly significant difference. There were n>30 eyes examined in each *Lmx1b*^V265D^ mutant group and n>20 eyes were examined in WT groups.

## Pyruvate administration

Ethyl pyruvate (Sigma-Aldrich, St. Louis, Missouri) was dissolved in the standard institutional drinking water to a dose of 2,000 mg/kg/day for adult mice based on the average volume mice consume. The mothers consumed this dose and delivered it to their suckling pups at an unknown dose throughout the first 3–4 weeks of life. At 4 weeks of age, the mice were weaned into separate cages. From 4 weeks on, the young mice consumed a dose of approximately 2,000 mg/kg/day based on their water consumption and body weight over weekly intervals. Untreated groups received the same drinking water without pyruvate. The water was changed once per week. Treatment was started on postnatal day 2. Births were checked daily between 9 am and 12 pm to determine the pups' age.

## Optic nerve assessment

We analyzed *Lmx1b*^V265D/+^ and WT control optic nerves for glaucomatous damage at 6 months of age (5.3–5.9 months range; majority are 5.8–5.9 months; sex balanced). Intracranial portions of optic nerves were dissected, processed, and analyzed as previously described (*Williams et al., 2017b*; *Howell et al., 2007*; *Nair et al., 2016*). Briefly, optic nerve cross-sections were stained with para-phenylenediamine (PPD) and examined for glaucomatous damage. PPD stains all myelin sheaths, but differentially darkly stains the myelin sheaths and the axoplasm of damaged axons. This allows for the sensitive detection and quantification of axon damage and loss. Optic nerves were prepared for analysis with a 48 hour fixation in 0.8% paraformaldehyde and 1.2% glutaraldehyde in 0.08 M phosphate buffer (pH 7.4) at 4 °C followed by overnight post-fix in osmium tetroxide at 4 °C. Nerves were washed twice for 10 minutes in 0.1 M phosphate buffer, once in 0.1 M sodium-acetate buffer, and dehydrated in graded ethanol concentrations. Tissues were then embedded in Embed 812 resin (Electron Microscopy Sciences, Fort Washington, PA, USA), and 1 μm-thick sections were stained in 1% PPD for ~40 minutes. Stained sections were compared using a damage scale that is validated against axon counting (*Howell et al., 2007*; *Howell et al., 2012*). Multiple sections of each nerve were considered when determining damage level. Nerves were determined to have one of 4 damage levels: (1) No glaucoma (NOE) – less than 5% axons damaged. This level of damage is seen in age- and sex-matched non-glaucomatous mice and is not due to glaucoma. We named this level no or early stage as some have early molecular changes when assessed with transcriptomics, but they cannot be distinguished from control by morphology; (*Williams et al., 2017a*; *Howell et al., 2011*). (2) Moderate damage (MOD) – average of 30% axon loss; (3) Severe (SEV) – greater than 50% axonal loss and extensive axon damage; and (4) Very severe (V. SEV) – glial scar over the vast majority of nerve with few remaining axons. Groups were compared statistically by Fisher's exact test. There were n=38 (untreated) and n=41 (pyruvate treated) *Lmx1b*^V265D/+^ mutant nerves examined. Cohort sizes for all *Lmx1b*^V265D/+^ mutant groups were determined using power calculations based on previous nerve damage data, ($\alpha$=0.05, power = 0.8). Sample sizes required to reach those power and $\alpha$ levels for optic nerve analysis determined the cohort sizes as optic nerve analysis required the largest sample sizes.

## Acknowledgements

The authors would like to thank the participants who contributed their data to the UK Biobank study for this study. The authors would also like to thank the members of the Simon John Laboratory for experimental and technical assistance and the Institute of Comparative Medicine for animal and veterinary care. This work is supported by NEI R01 015473 (LRP), NEI R01 032559 (LRP, JLW, AVS), EY036460 (LRP, JHK), NEI P30 EY014104 (JLW, AVS), NEI R01EY031424 (AVS), NIGMS R35-GM124836 (RD), NEI EY032507, EY032062, and EY018606 (SWMJ), an unrestricted Challenge Grant from Research to Prevent Blindness (NYC), and The Glaucoma Foundation (NYC). Further funding was provided by startup funds at Columbia University including the Precision Medicine Initiative, and by the New York Fund for Innovation in Research and Scientific Talent (NYFIRST; EMPIRE CU19-2660; SWMJ). This work is also supported by K23 (1K23EY032634) and the Research to Prevent Blindness Career Development Award (NZ), a Vision Core grant P30EY019007 (Columbia University) an unrestricted departmental

award from Research to Prevent Blindness (Columbia University). Supported by Fight for Sight (UK) (1956 A), and The Desmond Foundation (KVS); UK Research and Innovation Future Leaders Fellowship (MR/T040912/1), Alcon Research Institute Young Investigator Award and a Lister Institute of Preventative Medicine Fellowship (APK); financial support from the UK Department of Health through an award made by the National Institute for Health Research (NIHR) to Moorfields Eye Hospital National Health Service (NHS) Foundation Trust and University College London (UCL) Institute of Ophthalmology for a Biomedical Research Centre (BRC) for Ophthalmology (APK). The sponsor or funding organizations had no role in the design or conduct of this research. The content is solely the responsibility of the authors and does not necessarily represent the official views of the National Institutes of Health.

## Additional information

### Competing interests

Nicholas Tolman: inventor on patents/patent applications related to the use of ethyl pyruvate to treat glaucoma entitled "Pyruvate and Related Energetic Metabolites Modulate Resilience Against High Genetic Risk for Glaucoma" (U.S. 63/733985) and "Prevention and Treatment of Conditions Using Ethyl Pyruvate" (PCT/US24/20798). Neeru A Vallabh: has acted as a paid consultant or lecturer to Santen, Thea, NovaEye, Glaukos, and Elios. Nazlee Zebardast: reports consulting fees from Character Biosciences. Ron Do: reports being a scientific co-founder, consultant, and equity holder for Pensieve Health (pending) and being a consultant for Variant Bio and Character Bio. Anthony P Khawaja: has acted as a paid consultant or lecturer to Abbvie, Aerie, Allergan, Google Health, Heidelberg Engineering, Novartis, Reichert, Santen, Thea and Topcon. Janey L Wiggs: reports consulting fees from Editas and CRISPR Therapeutics. Simon WM John: founder of Myco Advising LLC, and Qura Inc and advises Starlight Bio Inc with no overlap to the currently presented research. Inventor on patents/patent applications related to the use of ethyl pyruvate to treat glaucoma entitled "Pyruvate and Related Energetic Metabolites Modulate Resilience Against High Genetic Risk for Glaucoma" (U.S. 63/733985) and "Prevention and Treatment of Conditions Using Ethyl Pyruvate" (PCT/US24/20798). Louis R Pasquale: reports consulting honoraria from Twenty Twenty. The other authors declare that no competing interests exist.

### Funding

| Funder | Grant reference number | Author |
|---|---|---|
| National Eye Institute | EY036460 | Jae H Kang<br>Louis R Pasquale |
| National Eye Institute | EY018606 | Simon WM John |
| National Eye Institute | R01EY031424 | Ayellet V Segrè |
| National Eye Institute | R01 015473 | Louis R Pasquale |
| National Eye Institute | R01 032559 | Ayellet V Segrè<br>Janey L Wiggs<br>Louis R Pasquale |
| National Eye Institute | EY014104 | Ayellet V Segrè<br>Janey L Wiggs |
| National Institute of General Medical Sciences | R35-GM124836 | Ron Do |
| National Eye Institute | EY032507 | Simon WM John |
| National Eye Institute | EY032062 | Simon WM John |
| Empire State Development | New York Fund for Innovation in Research and Scientific Talent EMPIRE CU19-2660 | Simon WM John |

| Funder | Grant reference number | Author |
|---|---|---|
| National Institute of Diabetes and Digestive and Kidney Diseases | 1K23EY032634 | Nazlee Zebardast |
| Research to Prevent Blindness Career Development Award | | Nazlee Zebardast |
| Columbia University | Vision Core P30EY019007 | Simon WM John |

The funders had no role in study design, data collection and interpretation, or the decision to submit the work for publication.

## Author contributions
Keva Li, Conceptualization, Formal analysis, Investigation, Visualization, Methodology, Writing – original draft, Writing – review and editing; Nicholas Tolman, Conceptualization, Data curation, Formal analysis, Validation, Investigation, Visualization, Methodology, Writing – original draft, Writing – review and editing; Ayellet V Segrè, Investigation, Methodology, Writing – review and editing; Kelsey V Stuart, Resources, Data curation, Investigation, Methodology, Writing – review and editing; Oana A Zeleznik, Neeru A Vallabh, Kuang Hu, Nazlee Zebardast, Akiko Hanyuda, Yoshihiko Raita, Christa Montgomery, Chi Zhang, Ron Do, Janey L Wiggs, Investigation, Writing – review and editing; Pirro G Hysi, Resources, Investigation, Methodology, Project administration, Writing – review and editing; Anthony P Khawaja, Conceptualization, Resources, Data curation, Supervision, Funding acquisition, Investigation, Visualization, Methodology, Project administration, Writing – review and editing; Jae H Kang, Louis R Pasquale, Conceptualization, Resources, Supervision, Funding acquisition, Validation, Investigation, Visualization, Methodology, Project administration, Writing – review and editing; Simon WM John, Conceptualization, Resources, Data curation, Formal analysis, Supervision, Funding acquisition, Validation, Investigation, Visualization, Methodology, Project administration, Writing – review and editing; UK Biobank Eye and Vision Consortium, Conceptualization, Data curation, Methodology, Project administration, Funding acquisition (Peng Khaw)

## Author ORCIDs
Keva Li (iD) https://orcid.org/0000-0001-5922-4955
Christa Montgomery (iD) https://orcid.org/0009-0004-2430-2772
Ron Do (iD) https://orcid.org/0000-0002-3144-3627
Jae H Kang (iD) https://orcid.org/0000-0003-4812-0557
Louis R Pasquale (iD) https://orcid.org/0000-0002-5835-3496

## Ethics
The UKBB study received approval from the National Health Service North West Multicentre Research Ethics Committee (reference number 06/MRE08/65) and the National Information Governance Board for Health and Social Care. This research, conducted under UK Biobank application number 36741, adhered to the tenets of the Declaration of Helsinki. I confirm that all necessary patient/participant consent has been obtained and the appropriate institutional forms have been archived, and that any patient/participant/sample identifiers included were not known to anyone (e.g., hospital staff, patients or participants themselves) outside the research group so cannot be used to identify individuals.

All mice were treated per the Association for Research in Vision and Ophthalmology's statement on the use of animals in ophthalmic research. The Institutional Animal Care and Use Committee of Columbia University approved all experimental protocols performed (protocol IDs: AC-AABE9554 and AC-AABD0557).

Reviewer #1 (Public review): https://doi.org/10.7554/eLife.105576.3.sa1
Reviewer #2 (Public review): https://doi.org/10.7554/eLife.105576.3.sa2
Author response https://doi.org/10.7554/eLife.105576.3.sa3

## Additional files

**Supplementary files**
MDAR checklist

**Data availability**
Data from the UK Biobank cannot be shared due to our Material Transfer Agreement. Individual-level data from the UK Biobank were accessed under application number 36741. Researchers may obtain access to these datasets by applying directly to each cohort: UK Biobank (https://www.ukbiobank.ac.uk/). To request access to the UK Biobank data, please make requests directly to the UK Biobank via https://www.ukbiobank.ac.uk/enable-your-research/apply-for-access. The glaucoma polygenic scores summary statistics from the MTAG analysis used in this study are available upon request to the corresponding authors of *Craig et al., 2020* and can be accessed at https://xikunhan.github.io/site/publication/. The code used in this study is available on GitHub (copy archived at *Li et al., 2025*).

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

## Appendix 1

### Supplementary methods

A total of 168 metabolites were quantified using nuclear magnetic resonance (NMR) spectroscopy. The panel includes Total Cholesterol, Total Cholesterol Minus HDL-C, Remnant Cholesterol (Non-HDL, Non-LDL Cholesterol), VLDL Cholesterol, Clinical LDL Cholesterol, LDL Cholesterol, HDL Cholesterol, Total Triglycerides, Triglycerides in VLDL, Triglycerides in LDL, Triglycerides in HDL, Total Phospholipids in Lipoprotein Particles, Phospholipids in VLDL, Phospholipids in LDL, Phospholipids in HDL, Total Esterified Cholesterol, Cholesteryl Esters in VLDL, Cholesteryl Esters in LDL, Cholesteryl Esters in HDL, Total Free Cholesterol, Free Cholesterol in VLDL, Free Cholesterol in LDL, Free Cholesterol in HDL, Total Lipids in Lipoprotein Particles, Total Lipids in VLDL, Total Lipids in LDL, Total Lipids in HDL, Total Concentration of Lipoprotein Particles, Concentration of VLDL Particles, Concentration of LDL Particles, Concentration of HDL Particles, Average Diameter for VLDL Particles, Average Diameter for LDL Particles, Average Diameter for HDL Particles, Phosphoglycerides, Total Cholines, Phosphatidylcholines, Sphingomyelins, Apolipoprotein B, Apolipoprotein A1, Total Fatty Acids, Degree of Unsaturation, Omega-3 Fatty Acids, Omega-6 Fatty Acids, Polyunsaturated Fatty Acids, Monounsaturated Fatty Acids, Saturated Fatty Acids, Linoleic Acid, Docosahexaenoic Acid, Alanine, Glutamine, Glycine, Histidine, Total Concentration of Branched-Chain Amino Acids (Leucine + Isoleucine + Valine), Isoleucine, Leucine, Valine, Phenylalanine, Tyrosine, Glucose, Lactate, Pyruvate, Citrate, 3-Hydroxybutyrate, Acetate, Acetoacetate, Acetone, Creatinine, Albumin, Glycoprotein Acetyls, Concentration of Chylomicrons and Extremely Large VLDL Particles, Total Lipids in Chylomicrons and Extremely Large VLDL, Phospholipids in Chylomicrons and Extremely Large VLDL, Cholesterol in Chylomicrons and Extremely Large VLDL, Cholesteryl Esters in Chylomicrons and Extremely Large VLDL, Free Cholesterol in Chylomicrons and Extremely Large VLDL, Triglycerides in Chylomicrons and Extremely Large VLDL, Concentration of Very Large VLDL Particles, Total Lipids in Very Large VLDL, Phospholipids in Very Large VLDL, Cholesterol in Very Large VLDL, Cholesteryl Esters in Very Large VLDL, Free Cholesterol in Very Large VLDL, Triglycerides in Very Large VLDL, Concentration of Large VLDL Particles, Total Lipids in Large VLDL, Phospholipids in Large VLDL, Cholesterol in Large VLDL, Cholesteryl Esters in Large VLDL, Free Cholesterol in Large VLDL, Triglycerides in Large VLDL, Concentration of Medium VLDL Particles, Total Lipids in Medium VLDL, Phospholipids in Medium VLDL, Cholesterol in Medium VLDL, Cholesteryl Esters in Medium VLDL, Free Cholesterol in Medium VLDL, Triglycerides in Medium VLDL, Concentration of Small VLDL Particles, Total Lipids in Small VLDL, Phospholipids in Small VLDL, Cholesterol in Small VLDL, Cholesteryl Esters in Small VLDL, Free Cholesterol in Small VLDL, Triglycerides in Small VLDL, Concentration of Very Small VLDL Particles, Total Lipids in Very Small VLDL, Phospholipids in Very Small VLDL, Cholesterol in Very Small VLDL, Cholesteryl Esters in Very Small VLDL, Free Cholesterol in Very Small VLDL, Triglycerides in Very Small VLDL, Concentration of IDL Particles, Total Lipids in IDL, Phospholipids in IDL, Cholesterol in IDL, Cholesteryl Esters in IDL, Free Cholesterol in IDL, Triglycerides in IDL, Concentration of Large LDL Particles, Total Lipids in Large LDL, Phospholipids in Large LDL, Cholesterol in Large LDL, Cholesteryl Esters in Large LDL, Free Cholesterol in Large LDL, Triglycerides in Large LDL, Concentration of Medium LDL Particles, Total Lipids in Medium LDL, Phospholipids in Medium LDL, Cholesterol in Medium LDL, Cholesteryl Esters in Medium LDL, Free Cholesterol in Medium LDL, Triglycerides in Medium LDL, Concentration of Small LDL Particles, Total Lipids in Small LDL, Phospholipids in Small LDL, Cholesterol in Small LDL, Cholesteryl Esters in Small LDL, Free Cholesterol in Small LDL, Triglycerides in Small LDL, Concentration of Very Large HDL Particles, Total Lipids in Very Large HDL, Phospholipids in Very Large HDL, Cholesterol in Very Large HDL, Cholesteryl Esters in Very Large HDL, Free Cholesterol in Very Large HDL, Triglycerides in Very Large HDL, Concentration of Large HDL Particles, Total Lipids in Large HDL, Phospholipids in Large HDL, Cholesterol in Large HDL, Cholesteryl Esters in Large HDL, Free Cholesterol in Large HDL, Triglycerides in Large HDL, Concentration of Medium HDL Particles, Total Lipids in Medium HDL, Phospholipids in Medium HDL, Cholesterol in Medium HDL, Cholesteryl Esters in Medium HDL, Free Cholesterol in Medium HDL, Triglycerides in Medium HDL, Concentration of Small HDL Particles, Total Lipids in Small HDL, Phospholipids in Small HDL, Cholesterol in Small HDL, Cholesteryl Esters in Small HDL, Free Cholesterol in Small HDL, and Triglycerides in Small HDL. A subset of 27 metabolites certified for broad diagnostic use by the European Union (EU) includes Total cholesterol, VLDL cholesterol, Clinical LDL cholesterol, HDL cholesterol, Total triglycerides, Apolipoprotein B, Apolipoprotein A1, Total fatty acids, Omega-3 fatty acids, Omega-6 fatty acids, Polyunsaturated fatty acids, Monounsaturated fatty acids, Saturated fatty acids, Docosahexaenoic acid, Alanine, Glycine, Histidine, Total concentration of branched- chain amino acids, Isoleucine, Leucine, Valine, Phenylalanine, Tyrosine, Glucose, Creatinine, Albumin, and Glycoprotein acetyls.

# Appendix 2

**Appendix 2—table 1.** Metabolic risk score beta (effect size) values for all UK Biobank participants (N=117,698).

| Metabolites | Beta-values |
|---|---|
| (Intercept) | –3.23 |
| Total Cholesterol | 548.50 |
| Cholesterol in Large HDL | 520.19 |
| Total Lipids in VLDL | 408.18 |
| Triglycerides in Chylomicrons and Extremely Large VLDL | 385.93 |
| Cholesterol in Chylomicrons and Extremely Large VLDL | 376.00 |
| Cholesteryl Esters in Very Large VLDL | 373.35 |
| Triglycerides in Very Large VLDL | 354.40 |
| Omega-6 Fatty Acids | 350.81 |
| Free Cholesterol in Large LDL | 343.95 |
| Phospholipids in VLDL | 326.74 |
| Free Cholesterol in Very Large VLDL | 319.33 |
| Total Lipids in HDL | 313.29 |
| Concentration of LDL Particles | 312.20 |
| Cholesterol in Large VLDL | 279.70 |
| Cholesteryl Esters in IDL | 258.77 |
| Cholesterol in Small LDL | 253.52 |
| Apolipoprotein B | 234.57 |
| Cholesteryl Esters in Very Small VLDL | 230.70 |
| Total Triglycerides | 225.33 |
| Total Cholesterol Minus HDL-C | 217.90 |
| Valine | 206.11 |
| Cholesterol in Very Large HDL | 197.87 |
| Triglycerides in HDL | 191.35 |
| Phospholipids in Small LDL | 191.15 |
| Cholesterol in Medium HDL | 185.32 |
| Phospholipids in Large LDL | 178.33 |
| Phospholipids in HDL | 177.36 |
| Cholesteryl Esters in Small VLDL | 175.91 |
| LDL Cholesterol | 162.05 |
| Total Lipids in Medium HDL | 145.96 |
| Free Cholesterol in Medium LDL | 142.98 |
| Total Fatty Acids | 140.65 |
| Leucine | 135.88 |
| Triglycerides in Large VLDL | 132.48 |
| Cholesteryl Esters in Large LDL | 131.14 |

*Appendix 2—table 1 Continued on next page*

*Appendix 2—table 1 Continued*

| Metabolites | Beta-values |
| --- | --- |
| Remnant Cholesterol (Non-HDL, Non-LDL -Cholesterol) | 125.15 |
| Cholesteryl Esters in Medium VLDL | 123.31 |
| Phospholipids in Large HDL | 118.13 |
| Omega-3 Fatty Acids | 113.37 |
| Cholesteryl Esters in Small LDL | 102.21 |
| Phospholipids in Very Large VLDL | 97.43 |
| Free Cholesterol in HDL | 96.54 |
| Triglycerides in VLDL | 95.51 |
| Free Cholesterol in IDL | 91.57 |
| Phospholipids in Very Large HDL | 91.28 |
| Free Cholesterol in Small LDL | 87.41 |
| Isoleucine | 85.64 |
| Phospholipids in Chylomicrons and Extremely Large VLDL | 72.54 |
| Phospholipids in Medium LDL | 67.41 |
| Triglycerides in Very Small VLDL | 66.01 |
| VLDL Cholesterol | 63.40 |
| Phospholipids in Very Small VLDL | 62.51 |
| Triglycerides in Small LDL | 61.15 |
| Triglycerides in Large LDL | 59.97 |
| Free Cholesterol in VLDL | 54.83 |
| Free Cholesterol in Very Small VLDL | 53.44 |
| Cholesteryl Esters in Medium LDL | 50.65 |
| Phospholipids in IDL | 48.84 |
| Cholesterol in Large LDL | 44.57 |
| Free Cholesterol in Small VLDL | 43.22 |
| Concentration of Very Small VLDL Particles | 38.34 |
| Concentration of Small VLDL Particles | 36.53 |
| Cholesteryl Esters in HDL | 33.26 |
| Concentration of Medium VLDL Particles | 33.22 |
| Phospholipids in Large VLDL | 32.50 |
| Concentration of Small HDL Particles | 29.93 |
| Cholesteryl Esters in Small HDL | 28.19 |
| Total Lipids in Small HDL | 25.46 |
| Cholesteryl Esters in LDL | 21.34 |
| Concentration of Large HDL Particles | 18.15 |
| Triglycerides in Medium LDL | 18.09 |
| Concentration of Medium HDL Particles | 17.63 |
| Concentration of Large VLDL Particles | 15.77 |

*Appendix 2—table 1 Continued on next page*

*Appendix 2—table 1 Continued*

| Metabolites | Beta-values |
|---|---|
| Free Cholesterol in Medium VLDL | 7.93 |
| Concentration of Very Large VLDL Particles | 6.13 |
| Concentration of Chylomicrons and Extremely Large VLDL Particles | 5.75 |
| Total Lipids in Medium LDL | 3.27 |
| Phosphoglycerides | 2.76 |
| Triglycerides in Small VLDL | 2.13 |
| Apolipoprotein A1 | 0.91 |
| Concentration of Very Large HDL Particles | 0.49 |
| Sphingomyelins | 0.15 |
| Linoleic Acid | 0.10 |
| Glutamine | 0.087 |
| Glycoprotein Acetyls | 0.085 |
| Average Diameter for VLDL Particles | 0.082 |
| Acetoacetate | 0.047 |
| Average Diameter for HDL Particles | 0.039 |
| Acetone | 0.036 |
| Tyrosine | 0.033 |
| Glycine | 0.030 |
| Degree of Unsaturation | 0.022 |
| Docosahexaenoic Acid | 0.012 |
| Glucose | 0.009 |
| 3-Hydroxybutyrate | −0.004 |
| Albumin | −0.006 |
| Phenylalanine | −0.008 |
| Creatinine | −0.016 |
| Histidine | −0.029 |
| Pyruvate | −0.057 |
| Citrate | −0.073 |
| Alanine | −0.083 |
| Lactate | −0.10 |
| Average Diameter for LDL Particles | −0.15 |
| Acetate | −0.21 |
| Phosphatidylcholines | −0.81 |
| Clinical LDL Cholesterol | −0.90 |
| Total Cholines | −2.18 |
| Concentration of HDL Particles | −6.05 |
| Triglycerides in IDL | −6.10 |
| Triglycerides in Very Large HDL | −8.37 |

*Appendix 2—table 1 Continued on next page*

*Appendix 2—table 1 Continued*

| Metabolites | Beta-values |
| --- | --- |
| Free Cholesterol in Small HDL | −10.78 |
| HDL Cholesterol | −13.62 |
| Phospholipids in Small VLDL | −19.05 |
| Triglycerides in Medium VLDL | −19.15 |
| Total Lipids in IDL | −25.11 |
| Cholesterol in Very Small VLDL | −30.03 |
| Free Cholesterol in Very Large HDL | −30.42 |
| Cholesterol in Medium VLDL | −32.45 |
| Phospholipids in Medium VLDL | −34.88 |
| Cholesteryl Esters in Large VLDL | −37.16 |
| Cholesterol in Small HDL | −38.44 |
| Cholesterol in Medium LDL | −39.45 |
| Concentration of IDL Particles | −42.85 |
| Triglycerides in Large HDL | −43.16 |
| Monounsaturated Fatty Acids | −48.49 |
| Phospholipids in Small HDL | −48.88 |
| Total Concentration of Lipoprotein Particles | −53.11 |
| Saturated Fatty Acids | −56.20 |
| Concentration of Small LDL Particles | −62.36 |
| Cholesteryl Esters in Chylomicrons and Extremely Large VLDL | −71.85 |
| Triglycerides in LDL | −78.07 |
| Free Cholesterol in Medium HDL | −78.70 |
| Cholesteryl Esters in Very Large HDL | −86.11 |
| Total Lipids in Lipoprotein Particles | −88.30 |
| Triglycerides in Small HDL | −89.86 |
| Phospholipids in LDL | −97.47 |
| Phospholipids in Medium HDL | −103.39 |
| Triglycerides in Medium HDL | −103.93 |
| Free Cholesterol in Large HDL | −110.51 |
| Free Cholesterol in Chylomicrons and Extremely Large VLDL | −114.90 |
| Free Cholesterol in Large VLDL | −115.30 |
| Cholesterol in Small VLDL | −119.66 |
| Concentration of Medium LDL Particles | −126.33 |
| Total Lipids in Small VLDL | −139.47 |
| Concentration of VLDL Particles | −148.00 |
| Total Lipids in LDL | −155.83 |
| Total Lipids in Medium VLDL | −175.12 |
| Total Lipids in Very Large HDL | −195.68 |

*Appendix 2—table 1 Continued on next page*

*Appendix 2—table 1 Continued*

| Metabolites | Beta-values |
| --- | --- |
| Total Lipids in Large LDL | −197.23 |
| Cholesteryl Esters in Medium HDL | −198.31 |
| Cholesteryl Esters in Large HDL | −234.91 |
| Total Phospholipids in Lipoprotein Particles | −286.84 |
| Concentration of Large LDL Particles | −294.73 |
| Cholesterol in IDL | −300.70 |
| Total Free Cholesterol | −331.88 |
| Cholesteryl Esters in VLDL | −359.42 |
| Total Lipids in Very Small VLDL | −360.23 |
| Total Lipids in Large HDL | −375.09 |
| Total Concentration of Branched-Chain Amino Acids (Leucine + Isoleucine + Valine) | −411.21 |
| Total Lipids in Large VLDL | −432.35 |
| Free Cholesterol in LDL | −454.32 |
| Polyunsaturated Fatty Acids | −460.67 |
| Cholesterol in Very Large VLDL | −518.35 |
| Total Lipids in Small LDL | −561.42 |
| Total Lipids in Very Large VLDL | −716.90 |
| Total Lipids in Chylomicrons and Extremely Large VLDL | −799.17 |
| Total Esterified Cholesterol | −838.60 |

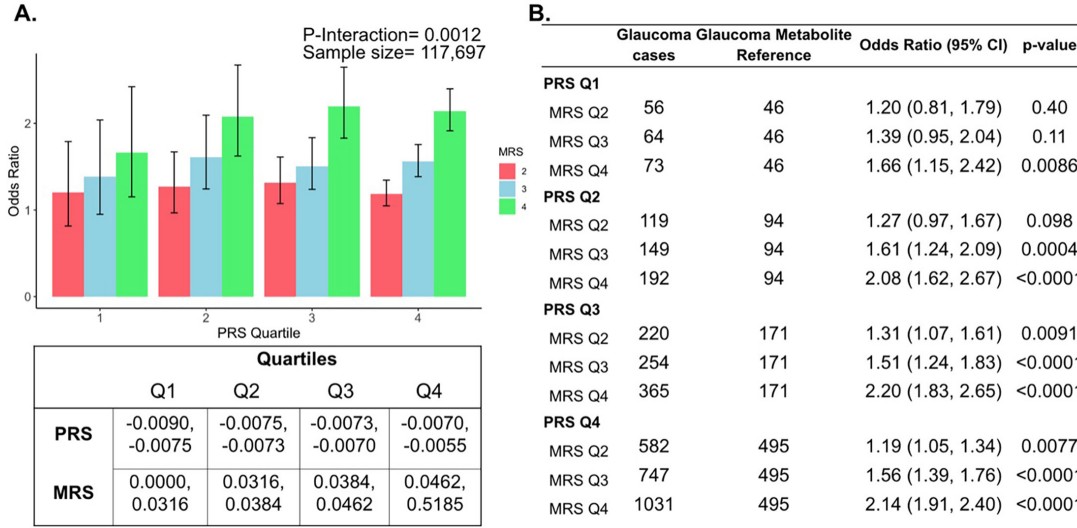

**Appendix 2—figure 1.** Interaction of the holistic metabolite risk score (n=168 metabolites) and polygenic risk score (PRS) on glaucoma risk. (**A**) The bar chart shows the interaction of holistic probit-transformed metabolite risk score (MRS) with glaucoma genetic predisposition in each PRS quartile. In each glaucoma PRS quartile, the lowest metabolic sum quartile (Q1) is the metabolite reference group used to calculate the odds ratios. Each color represents the MRS quartiles (red = second quartile; blue = third quartile; and green = fourth quartile). Error bars show 95% confidence interval (CI). The table under the bar chart shows the ranges for the PRS and MRS quartiles. (**B**) Table showing odds ratios for glaucoma by polygenic risk score (PRS) and MRS within various quartiles. The number of glaucoma cases within each MRS quartile and the number of glaucoma cases in the first quartile of MRS (Q1), labeled as glaucoma metabolite reference, are used to calculate the odds ratios. This analysis is adjusted for time since the last meal/drink (hours), age (years), age-squared (years-squared), sex, ethnicity (Asian, Black, White,

*Appendix 2—figure 1 continued on next page*

*Appendix 2—figure 1 continued*
and other), season, time of day of specimen collection (morning, afternoon, night), smoking status (never, past, and current smoker), alcohol intake, caffeine intake, physical activity (metabolic equivalent of task [MET] hours/week), body mass index (kg/m$^2$), average systolic blood pressure (mm Hg), history of diabetes (yes or no), HbA1c (mmol/mol), history of coronary artery disease, systemic beta-blocker use, oral steroid use, and spherical equivalent refractive error (diopters).

**Appendix 2—table 2.** Unadjusted levels of plasma metabolites lactate, pyruvate, and citrate in UK Biobank participants and among the top 10% of glaucoma polygenic risk score (PRS).

| Characteristic | Total population | | | | Top 10% PRS | | | |
| --- | --- | --- | --- | --- | --- | --- | --- | --- |
| | Overall | No Glaucoma | Glaucoma | p-value | Overall | No Glaucoma | Glaucoma | p-value |
| Sample size | 117,698 | 113,040 | 4,658 | | 11,770 | 10,077 | 1,693 | |
| Lactate, mmol/l (median [IQR]) | 3.94 [3.24, 4.75] | 3.95 [3.24, 4.76] | 3.82 [3.17, 4.60] | <0.001 | 3.94 [3.25, 4.75] | 3.96 [3.27, 4.78] | 3.79 [3.16, 4.58] | <0.001 |
| Pyruvate, mmol/l (median [IQR]) | 0.080 [0.063, 0.10] | 0.080 [0.06, 0.10] | 0.0780 [0.06, 0.10] | <0.001 | 0.080 [0.06, 0.10] | 0.080 [0.06, 0.10] | 0.077 [0.06, 0.09] | <0.001 |
| Citrate, mmol/l (median [IQR]) | 0.065 [0.057, 0.074] | 0.065 [0.06, 0.07] | 0.065 [0.06, 0.07] | 0.51 | 0.065 [0.06, 0.07] | 0.06 [0.06, 0.07] | 0.07 [0.06, 0.07] | 0.87 |

Abbreviations IQR, interquartile range.

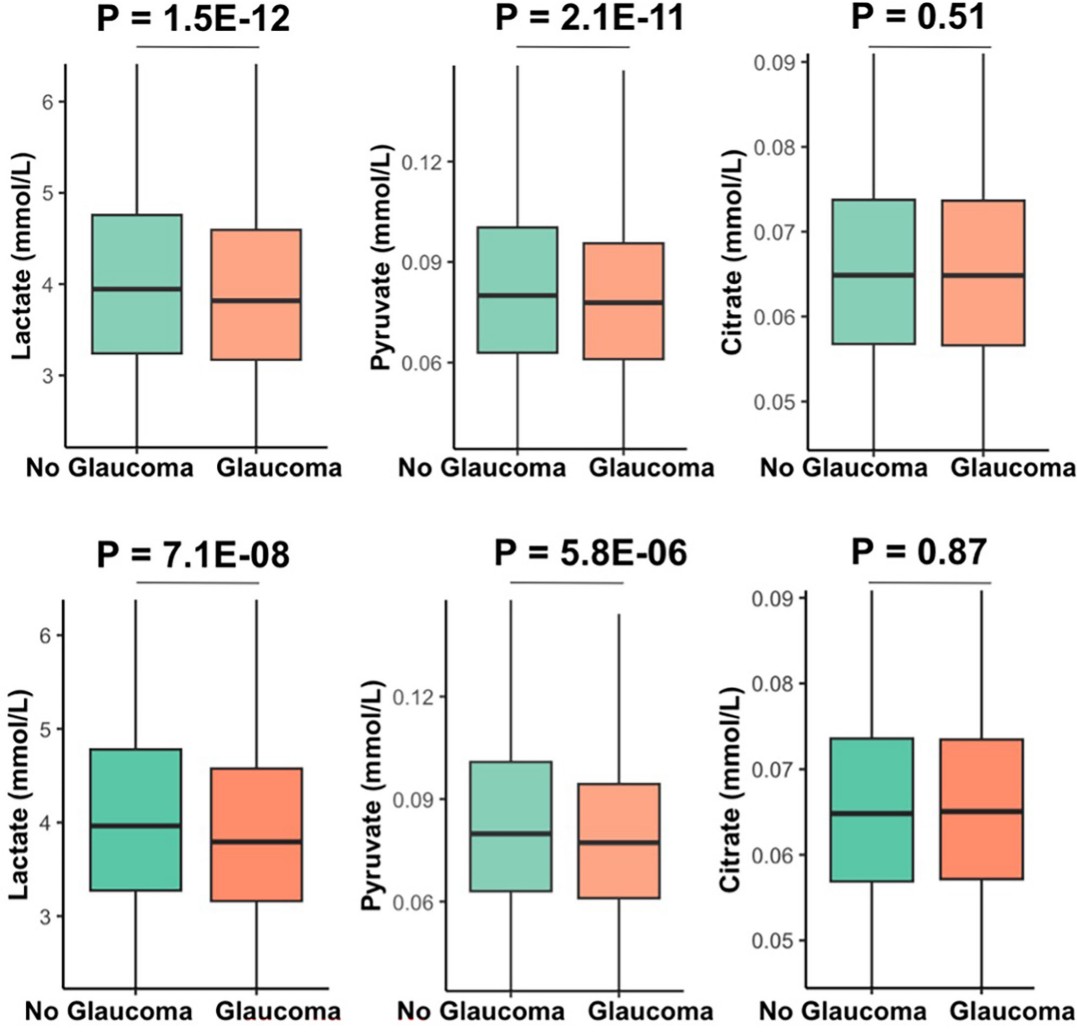

**Appendix 2—figure 2.** Unadjusted levels of plasma metabolites lactate, pyruvate, and citrate in all UK Biobank participants (top row) and among the top 10% of glaucoma polygenic risk score (bottom row).

