## [Editor Report · eLife Assessment]

This study presents a **valuable** finding on the importance of the plasma metabolome in glaucoma risk prediction. The evidence supporting the claims of the authors is **solid** and the work offers insights for the design of protective therapeutic strategies for glaucoma. The authors have addressed the concerns of the reviewers and reported on the limitations of the study.

---

## [Referee Report · Reviewer #1 (Public review)]

Summary:

The Authors explore associations between plasma metabolites and glaucoma, a primary cause of irreversible vision loss worldwide. The study relies on measurements of 168 plasma metabolites in 4,658 glaucoma patients and 113,040 controls from the UK Biobank. The Authors show that metabolites improve the prediction of glaucoma risk based on polygenic risk score (PRS) alone, albeit weakly. The Authors also report a "metabolomic signature" that is associated with a reduced risk (or "resilience") for developing glaucoma among individuals in the highest PRS decile (reduction of risk by an estimated 29%). The Authors highlight the protective effect of pyruvate, a product of glycolysis, for glaucoma development and show that this molecule mitigates elevated intraocular pressure and optic nerve damage in a mouse model of this disease.

Strengths:

This work provides additional evidence that glycolysis may play a role in the pathophysiology of glaucoma. Previous studies have demonstrated the existence of an inverse relationship between intraocular pressure and retinal pyruvate levels in animal models (Hader et al. 2020, PNAS 117(52)) and pyruvate supplementation is currently being explored for neuro-enhancement in patients with glaucoma (De Moraes et al. 2022, JAMA Ophthalmology 140(1)). The study design is rigorous and relies on validated standard methods. Additional insights gained from a mouse model are valuable.

Weaknesses:

Caution is warranted when examining and interpreting the results of this study. Among all participants (cases and controls) glaucoma status was self-reported, determined on the basis of ICD codes or previous glaucoma laser/surgical therapy. This is problematic as it is not uncommon for individuals in the highest PRS decile to have undiagnosed glaucoma (as shown in previous work by some of the authors of this article). The Authors acknowledge a "relatively low glaucoma prevalence in the highest decile group" but do not explore how undiagnosed glaucoma may affect their results. This also applies to all controls selected for this study. The Authors state that "50 to 70% of people affected [with glaucoma] remain undiagnosed". Therefore, the absence of self-reported glaucoma does not necessarily indicate that the disease is not present. Validation of the findings from this study in humans is, therefore, critical. This should ideally be performed in a well-characterized glaucoma cohort, in which case and control status has been assessed by qualified clinicians.

The authors indicate that within the top decile of PRS participants with glaucoma are more likely to be of white ethnicity, while they are more likely to be of Black and Asian ethnicity if they are in the bottom half of PRS. Have the Authors explored how sensitive their predictions are to ethnicity? Since their cohort is predominantly of European ancestry (85.8%), would it make sense to exclude other ethnicities to increase the homogeneity of the cohort and reduce the risk for confounders that may not be explicitly accounted for?

The authors discuss the importance of pyruvate, and lactate for retinal ganglion cell survival along with that of several lipoproteins for neuroprotection. However, there is a distinction to be made between locally produced/available glycolysis end products and lipoproteins and those circulating in the blood. It may be useful to discuss this in the manuscript, and for the Authors to explore if plasma metabolites may be linked to metabolism that takes place past the blood-retinal barrier.

Comments on revisions:

The authors have addressed all of my concerns.

---

## [Referee Report · Reviewer #2 (Public review)]

Summary

The authors have used the UK Biobank data to interrogate the association between plasma metabolites and glaucoma.

(1) They initially assessed plasma metabolites as predictors of glaucoma: The addition of NMR-derived metabolomic data to existing models containing clinical and genetic data was marginal.

(2) They then determined whether certain metabolites might protect against glaucoma in individuals at high genetic risk: Certain molecules in bioenergetic pathways (lactate, pyruvate and citrate) conferred protection.

(3) They provide support for protection conferred by pyruvate in a murine model.

Weaknesses

(1) Although it is an invaluable treasure trove of data, selection bias and self-reporting are inescapable problems when using the UK Biobank data for glaucoma research. The high-impact glaucoma-related GWAS publications (Ref 26 and 27) referenced in support of the method suffer the same limitations. This doesn't negate the conclusions but must be taken into consideration. The authors might note that it is somewhat reassuring that the proportion of glaucoma cases (4%) is close to what would be expected in a population-based study of 40-69-year-olds of predominantly white ethnicity.

(2) As noted by the authors, a limitation is the predominantly white ethnicity profile that comprises the UK Biobank.

(3) Also as noted by the authors, the study is cross-sectional and is limited by the "correlation does not imply causation" issue.

(4) The optimal collection, transport and processing of the samples for NMR metabolite analysis is critical for accurate results. Strict policies were in place for these procedures, but deviations from protocol remain an unknown influence on the data.

(5) In addition, all UK Biobank blood samples had unintended dilution during the initial sample storage process at UK Biobank facilities. Julkunen, H. et al. Atlas of plasma NMR biomarkers for health and disease in 118,461 individuals from the UK Biobank. Nat Commun 14, 604 (2023) Samples from aliquot 3, used for the NMR measurements, suffered from 5-10% dilution. (Allen, Naomi E., et al. Wellcome Open Research 5 (2021): 222.) Julkunen et al. report that "The dilution is believed to come from mixing of participant samples with water due to seals that failed to hold a system vacuum in the automated liquid handling systems. While this issue is likely to have an impact on some of the absolute biomarker concentration values, it is expected to have limited impact on most epidemiological analyses."

Strengths

The huge sample size supports a powerful statistical analysis and the opportunity for the inclusion of multiple covariates and interactions without overfitting the models.

The authors have constructed a robust methodology and statistical design.

The manuscript is well-written, and the study is logically presented.

The Figures are of good quality.

Broadly, the conclusions are justified by the findings.

Impact

The findings advance personalized prognostics for glaucoma that combine metabolomic and genetic data. In addition, the protective effect of certain metabolites influences further research on novel therapeutic strategies.

Comments on revisions:

The authors have thoughtfully and comprehensively addressed my comments. I have no further comments.

---

## [Author Response]

The following is the authors’ response to the original reviews

**Public Reviews:**

**Reviewer #1 (Public review):**
Summary:The authors explore associations between plasma metabolites and glaucoma, a primary cause of irreversible vision loss worldwide. The study relies on measurements of 168 plasma metabolites in 4,658 glaucoma patients and 113,040 controls from the UK Biobank. The authors show that metabolites improve the prediction of glaucoma risk based on polygenic risk score (PRS) alone, albeit weakly. The authors also report a "metabolomic signature" that is associated with a reduced risk (or "resilience") for developing glaucoma among individuals in the highest PRS decile (reduction of risk by an estimated 29%). The authors highlight the protective effect of pyruvate, a product of glycolysis, for glaucoma development and show that this molecule mitigates elevated intraocular pressure and optic nerve damage in a mouse model of this disease.Strengths:This work provides additional evidence that glycolysis may play a role in the pathophysiology of glaucoma. Previous studies have demonstrated the existence of an inverse relationship between intraocular pressure and retinal pyruvate levels in animal models (Hader et al. 2020, PNAS 117(52)) and pyruvate supplementation is currently being explored for neuro-enhancement in patients with glaucoma (De Moraes et al. 2022, JAMA Ophthalmology 140(1)). The study design is rigorous and relies on validated, standard methods. Additional insights gained from a mouse model are valuable.

We thank the reviewer for these supportive comments.

Weaknesses:Caution is warranted when examining and interpreting the results of this study. Among all participants (cases and controls) glaucoma status was self-reported, determined on the basis of ICD codes or previous glaucoma laser/surgical therapy. This is problematic as it is not uncommon for individuals in the highest PRS decile to have undiagnosed glaucoma (as shown in previous work by some of the authors of this article). The authors acknowledge a "relatively low glaucoma prevalence in the highest decile group" but do not explore how undiagnosed glaucoma may affect their results. This also applies to all controls selected for this study. The authors state that "50 to 70% of people affected [with glaucoma] remain undiagnosed". Therefore, the absence of self-reported glaucoma does not necessarily indicate that the disease is not present. Validation of the findings from this study in humans is, therefore, critical. This should ideally be performed in a well-characterized glaucoma cohort, in which case and control status has been assessed by qualified clinicians.

We appreciate the comment regarding the challenges of glaucoma ascertainment in UK Biobank. This is a valid limitation, as glaucoma in UK Biobank is based on self-reports and hospital records rather than comprehensive ophthalmologic examinations for all participants. To the best of our knowledge, there is no comparably sized dataset where all participants have undergone standardized glaucoma assessments, comprehensive metabolomic profiling, and high-throughput genotyping. Work is currently ongoing to link UK Biobank data to ophthalmic EMR data, which will help confirm self-reported diagnoses. This work is not complete, and the coverage of the cohort from such linkage is uncertain at present. Nonetheless, several factors speak to the validity of our findings. The top members of the metabolomic signature associated with resilience in the top decile of glaucoma polygenic risk score (PRS) decile—lactate (P=8.8E-12) and pyruvate (P=1.9E-10) —had robust values for statistical significance after appropriate adjustment for multiple comparisons, with additional validation for pyruvate in a human-relevant, glaucoma mouse model. Strikingly, the glaucoma odds ratio (OR) for subjects in the highest quartile of glaucoma PRS and metabolic risk score (MRS) was 25-fold, using participants in the lowest quartile of glaucoma PRS and MRS as the reference group. An effect size this large for a putative glaucoma determinant has only been seen for intraocular pressure (IOP), which is now largely accepted to be in the causal pathway of the disease.

The Discussion now contains the following statement: “A second limitation is that glaucoma ascertainment in the UK Biobank is based on self-reported diagnoses and hospital records rather than comprehensive ophthalmologic examinations. Nonetheless, it is reassuring that the prevalence of glaucoma in our sample (~4%) is similar to a directly performed disease burden estimate in a comparable, albeit slightly older, United Kingdom sample (2.7%)(79)”. (Lines 379-382)

The authors indicate that within the top decile of PRS participants with glaucoma are more likely to be of white ethnicity, while they are more likely to be of Black and Asian ethnicity if they are in the bottom half of PRS. Have the authors explored how sensitive their predictions are to ethnicity? Since their cohort is predominantly of European ancestry (85.8%), would it make sense to exclude other ethnicities to increase the homogeneity of the cohort and reduce the risk for confounders that may not be explicitly accounted for?

Comparing data in Tables 3 and 4 of the manuscript, we observe that, on a percentage basis, more individuals have glaucoma in the highest 10th percentile of risk compared to the lowest 50th percentile of risk across all ancestral groups. We recently reported that the risk of glaucoma increases with each standard deviation increase in the glaucoma PRS across ancestral groups in the UK Biobank, utilizing a slightly different sample size (see Author response table 1 below). (1) Since the PRS is applicable across ancestral groups, we aim to make our results as generalizable as possible; therefore, we prefer to report our findings for all ethnic groups and not restrict our results to Europeans.

**Author response table 1. sa3table1:** Performance of the mtGPRS Across Ancestral Groups in the UK Biobank.

Ancestry	N, glaucoma cases	N, controls	N, total	OR [95% CI]
African	184	2,448	2,632	1.25[0.97-1.60]
Asian	199	4,475	4,674	1.63[1.34-1.98]
UK European	7,973	162,190	170,163	2.84[2.73-2.92]
Other European	220	5,837	6,057	1.67[1.43-1.96]

Abbreviations: mtGPRS, multitrait analysis of GWAS polygenic risk score; OR, odds ratio; CI, confidence interval.

UK Biobank ancestry was genetically inferred based on principal component analysis. The OR represents the risk associated with each standard deviation change in mtGRS and is adjusted for multiple covariates including age, sex, and medical comorbidities.

In the discussion, we stated that “... we chose to analyze Europeans and non-Europeans together to make the results as generalizable as possible.” (Lines 378-379)

The authors discuss the importance of pyruvate, and lactate for retinal ganglion cell survival, along with that of several lipoproteins for neuroprotection. However, there is a distinction to be made between locally produced/available glycolysis end products and lipoproteins and those circulating in the blood. It may be useful to discuss this in the manuscript, and for the authors to explore if plasma metabolites may be linked to metabolism that takes place past the blood-retinal barrier.

As the reviewer points out, it is crucial to interpret the results for lipoproteins within the context of their access to the blood-retinal barrier. Even for smaller metabolites like pyruvate and lactate, it is essential to consider local production versus serum-derived molecules in mediating any neuroprotective effects. Our murine data suggest that exogenous pyruvate contributed to neuroprotection. However, for the other glycolysis-related metabolites (lactate and citrate), we cannot rule out the possibility that locally produced metabolites may also contribute to neuroprotection. None of the lipoproteins identified as potential resilience biomarkers had an adjusted P-value of less than 0.05. Nevertheless, HDL analytes can cross blood-ocular barriers to enter the aqueous humor.(2) Therefore, it is also possible for serum-derived HDL to influence retinal ganglion cell homeostasis. Overall, much more research is needed to clarify the roles of locally produced versus serum-derived factors in conferring resilience to genetic predisposition to glaucoma.

We have added the following sentences to the discussion:

“Notably, although our validation data confirm the neuroprotective effects of exogenous pyruvate, it remains possible that endogenously produced pyruvate within ocular tissues may also contribute to RGC protection.” (Lines 329-331)

“Furthermore, as HDL analytes can cross blood-ocular barriers,(78) there is a plausible route for serum-derived HDL to influence RGC homeostasis. Nonetheless, the relative contributions of circulating lipoproteins versus local synthesis within ocular tissues remain unclear and warrant further investigation.” (Lines 355-358)

“Incorporating ocular physiology and blood-retinal barrier considerations when interpreting lipoproteins as potential resilience biomarkers will be critical for future studies aimed at understanding and therapeutically targeting increased glaucoma risk.” (Lines 360-363)

**Reviewer #2 (Public review):**
SummaryThe authors have used the UK Biobank data to interrogate the association between plasma metabolites and glaucoma.(1) They initially assessed plasma metabolites as predictors of glaucoma: The addition of NMR-derived metabolomic data to existing models containing clinical and genetic data was marginal.(2) They then determined whether certain metabolites might protect against glaucoma in individuals at high genetic risk: Certain molecules in bioenergetic pathways (lactate, pyruvate, and citrate) conferred protection.(3) They provide support for protection conferred by pyruvate in a murine model.Strengths(1) The huge sample size supports a powerful statistical analysis and the opportunity for the inclusion of multiple covariates and interactions without overfitting the models.(2) The authors have constructed a robust methodology and statistical design.(3) The manuscript is well written, and the study is logically presented.(4) The figures are of good quality.(5) Broadly, the conclusions are justified by the findings.

We thank the reviewer for these supportive comments.

Weaknesses(1) Although it is an invaluable treasure trove of data, selection bias and self-reporting are inescapable problems when using the UK Biobank data for glaucoma research. The high-impact glaucoma-related GWAS publications (references 26 and 27) referenced in support of the method suffer the same limitations. This doesn't negate the conclusions but must be taken into consideration. The authors might note that it is somewhat reassuring that the proportion of glaucoma cases (4%) is close to what would be expected in a population-based study of 40-69-year-olds of predominantly white ethnicity.

While there are limitations when open-angle glaucoma (OAG) is ascertained by self-report, as discussed above, we agree with the reviewer that the prevalence of glaucoma is consistent with data from population-based studies of Europeans who are 40-69 years of age.

We also want to point out that references 26 and 27 indicate glaucoma self-reports can be an acceptable surrogate for OAG that is ascertained by clinical evaluation. Consider the methodologic details for each study:

Reference 26 is a 4-stage genome-wide meta-analysis to identify loci for OAG from 21 independent populations. The phenotypic definition of OAG was based on clinical assessment in the discovery stage, and 7286 glaucoma self-reports from the UK Biobank served as an effective replication set. It is also important to note that 120 out of the 127 discovered OAG loci were nominally replicated in 23andMe, where glaucoma was ascertained entirely by self-report.

Reference 27 is a genome-wide meta-analysis to identify IOP genetic loci, an important endophenotype for OAG. The study identified 112 loci for IOP. These loci were incorporated into a glaucoma prediction model in the NEIGHBORHOOD study and the UK Biobank. The area under the receiver operator curve was 0.76 and 0.74, respectively, in these studies. While the AUCs were similar, OAG was ascertained clinically in NEIGHBORHOOD and largely by self-report in UK Biobank.

Finally, a strength of the UK Biobank is that selection bias is minimized. Patients need not be insured or aligned to the study for any reason aside from being a UK resident. There is indeed a healthy bias in the UK Biobank. Ambulatory patients who tend to be health conscious and willing to donate their time and provide biological specimens tend to participate. We agree with the reviewer that the use of self-reported cases does not negate the conclusions, and hopefully, future iterations of the UK Biobank where clinical validation of self-reports are performed will confirm these findings, which already have some validation in a preclinical glaucoma model.

We add the following sentence to the first action item above regarding our case ascertainment method. “Nonetheless, it is reassuring that the prevalence of glaucoma in our sample (~4%) is similar to a directly performed disease burden estimate in a comparable, albeit slightly older, United Kingdom sample (2.7%)..”(3) (Lines 381-383)

(2) As noted by the authors, a limitation is the predominantly white ethnicity profile that comprises the UK Biobank.(3) Also as noted by the authors, the study is cross-sectional and is limited by the "correlation does not imply causation" issue.

While the epidemiological arm of our study was cross-sectional, the studies testing the ability of pyruvate to mitigate the glaucoma phenotype in mice with the *Lmxb1* mutation were prospective.

We already pointed out in the discussion that pyruvate supplementation reduced glaucoma incidence in a human-relevant genetic mouse model.

(4) The optimal collection, transport, and processing of the samples for NMR metabolite analysis is critical for accurate results. Strict policies were in place for these procedures, but deviations from protocol remain an unknown influence on the data.

Comments 4 and 5 are related and will be addressed after comment 5.

(5) In addition, all UK Biobank blood samples had unintended dilution during the initial sample storage process at UK Biobank facilities. Julkunen, H. et al. Atlas of plasma NMR biomarkers for health and disease in 118,461 individuals from the UK Biobank. Nat Commun 14, 604 (2023) Samples from aliquot 3, used for the NMR measurements, suffered from 5-10% dilution. (Allen, Naomi E., et al. Wellcome Open Research 5 (2021): 222.) Julkunen et al. report that "The dilution is believed to come from mixing of participant samples with water due to seals that failed to hold a system vacuum in the automated liquid handling systems. While this issue is likely to have an impact on some of the absolute biomarker concentration values, it is expected to have limited impact on most epidemiological analyses."

We thank the reviewer for making us aware of the unintended sample dilution issue from aliquot 3, used for NMR metabolomics in UK Biobank participants. While ~98% of samples experienced a 5-10% dilution, this would not affect our reported results, which did not rely on absolute biomarker concentrations. All metabolites in the main tables were probit transformed and used as continuous variables per 1 standard deviation increase. Nonetheless, in supplemental material, we show the unadjusted median levels of pyruvate (in mmol/L) were higher in participants without glaucoma vs those with glaucoma, both in the population overall and in those in the top 10 percentile of glaucoma risk.

Furthermore, we see no evidence in the literature that unidentified protocol deviations might impact metabolite results in UK Biobank participants. For example, a recent study evaluated the relationship between a weighted triglyceride-raising polygenic score (TG.PS) and type 3 hyperlipidemia (T3HL) in the Oxford Biobank (OBB) and the UK Biobank. In both biobanks, metabolomics was performed on the Nightingale NMR platform. A one standard deviation increase in TG.PS was associated with a 13% and 15.2% increased risk of T3HL in the OBB and UK Biobank, respectively.(4) Replication of the OBB result in the UK Biobank suggests there are no additional concerns regarding the processing of the UK Biobank for NMR metabolomics. Of course, we remain vigilant for protocol deviations that might call our results into question and will seek to validate our findings in other biobanks in future research.

ImpactThe findings advance personalized prognostics for glaucoma that combine metabolomic and genetic data. In addition, the protective effect of certain metabolites influences further research on novel therapeutic strategies.
**Recommendations for the authors:**

**Reviewer #1 (Recommendations for the authors):**
Given the uncertainty in the proportion of controls with undiagnosed glaucoma, it may be appropriate to include a sensitivity analysis in the manuscript. The authors could then provide the readers with an estimate of how sensitive their predictions are to the proportion of undiagnosed individuals among controls.

Since UK Biobank participants did not undergo standardized clinical assessments, it is not possible to perform sensitivity analyses as we don’t know which controls might have glaucoma, although we can offer the following comments.

We are performing a cross-sectional, prospective, detailed glaucoma assessment of participants in the top and bottom 10 percent of genetic predisposition recruited from BioMe at Icahn School of Medicine at Mount Sinai and Mass General Brigham Biobank at Harvard Medical School. We find that 21% of people in the top decile of genetic risk have glaucoma,(5) which compares reasonably well to the 15% of people in the top 10% of genetic risk in the UK Biobank. This underscores the assertion that our definition of glaucoma in the UK Biobank, while not ideal, is a reasonable surrogate for a detailed clinical assessment.

Currently, 10,077 subjects in the top decile of glaucoma genetic predisposition did not meet our definition of glaucoma. If we assume that the glaucoma prevalence is 3% and 50% of people with glaucoma are undiagnosed, then that would translate to an additional 150 cases misclassified as controls, which could either drive our result to the null, have no impact on our current result or contribute to a false positive result, depending on their pyruvate (and other metabolite) levels.

We have already addressed the issue of a lack of standardized exams in the UK Biobank and the need for more studies to confirm our findings.

**Reviewer #2 (Recommendations for the authors):**
(1) I am curious about the proposed reason for some individuals having metabolic profiles conferring resilience. Plasma pyruvate levels are normally distributed. Is it simply the case that some individuals with naturally high levels of pyruvate are fortuitously protected against glaucoma? Some sort of self-regulation mechanism seems unlikely.

Thank you for your insightful question regarding the potential mechanism underlying the association between pyruvate levels and glaucoma resilience. There may be modest inter-individual differences which can have significant physiological implications, particularly in the context of neurodegeneration and metabolic stress. One possibility is that individuals with naturally higher pyruvate levels may benefit from pyruvate's known neuroprotective and metabolic support functions(6–8), which could confer resilience against the oxidative and bioenergetic challenges associated with glaucoma. Pyruvate is important for cellular metabolism, redox balance, and mitochondrial function - processes that are increasingly implicated in glaucomatous neurodegeneration. (9)Elevated pyruvate levels support mitochondrial ATP production(10), buffer oxidative stress,(11) and impact metabolic flux(12,13) through pathways such as the tricarboxylic acid cycle and NAD+/NADH homeostasis. This is consistent with prior studies suggesting that mitochondrial dysfunction contributes to retinal ganglion cell vulnerability in glaucoma.(14–17) While a direct self-regulation mechanism may seem unlikely, both genetic and environmental factors can influence pyruvate metabolism, which could lead to subtle but clinically meaningful variations in its levels. Our findings are supported by validation in a mouse model, which suggests that the association is less likely fortuitous, but there may be an underlying biological process that merits further mechanistic investigation. Future studies incorporating longitudinal metabolic profiling and functional validation in human-derived models will help better understand this relationship.

(2) Conceivably, the higher levels of pyruvate and lactate may have resulted from recent exercise and may reflect individuals with high levels of exercise that confers resilience against glaucoma by independent mechanisms such as improved blood flow. Any way to rule that out from the UK Biobank data?

Thank you for raising this important point. To account for the potential confounding effects of physical activity, we adjusted for metabolic equivalents of task (METs) in our models, a standardized measure of physical activity available in the UK Biobank. By incorporating METs as a covariate, we aimed to minimize the influence of individual exercise levels on plasma pyruvate and lactate levels. This helps us ascertain that the observed associations are not solely attributable to differences in physical activity. However, we do acknowledge that longitudinal analysis of exercise patterns would provide further clarity on this relationship.

(3) It may be worth mentioning that the retinal ganglion cells contain a plasma membrane monocarboxylate transporter that supports pyruvate and lactate uptake from the extracellular space.

Thank you for this extremely insightful suggestion on retinal ganglion cell (RGC) expression of monocarboxylate transporters, which can facilitate the uptake of pyruvate and lactate from the extracellular space. This is relevant for our study, given the high metabolic demands of RGCs and their reliance on both glycolytic and oxidative metabolism for neuroprotection and survival.

We acknowledged this in the discussion section of the manuscript by adding the following statement: "RGCs express monocarboxylate transporters, which facilitate the uptake of extracellular pyruvate and lactate, improving energy homeostasis, neuronal metabolism, and survival.” (Lines 309-311)

(4) The mechanism of protection in the mice, at least in part, is likely due to the lower IOP in the pyruvate-treated animals. Did the authors investigate the influence of pyruvate on IOP in the UK Biobank data (about 110,000 individuals had IOP measurements)?

Thank you for your suggested investigation. We ran the suggested analysis among 68,761 individuals with IOP measurements and metabolomic profiling. Imputed pretreatment IOP values for participants using ocular hypotensive agents were calculated by dividing the measured IOP by 0.7, based on the mean IOP.

We plotted the relationship between IOP and pyruvate levels (probit transformed). We compared participants with pyruvate levels +2 standard deviations, above the mean (red dashed line), which has a probit-transformed value of 2 and an absolute concentration of 0.15 mmol/L. We found a statistically significant difference between the groups (p=0.017) using the Welch two-sample t-test. We have not added this analysis to the manuscript, but readers can find the data here as the reviews are public. We acknowledge and addressed the dilutional issue above, where we utilized probit-transformed metabolite levels analyzed as continuous variables per 1 SD increase, rather than absolute concentrations.

(5) Line 88: I suggest changing "patients" to "affected individuals". The term "patients" tends to imply that the individual has already been diagnosed, but the idea being conveyed is about underdiagnosis in the population.

Thank you for your suggestion.

We have added the change from "patients" to "affected individuals" in the introduction. (Line 90)

(6) Line 93: "However, glaucoma is also significantly affected by environmental and lifestyle factors,10-14". Although lifestyle risk factors such as caffeine intake, alcohol, smoking, and air pollution have been reported, the associations are generally weak and inconsistently reported. Consider modifying this notion to stress the emerging evidence around gene-environment interactions (reference 14) rather than environmental factors per se, with the implication that genes + metabolism may be greater than the sum of the parts.

Thank you for this thoughtful suggestion to highlight gene-environment interactions, where genetic susceptibility may amplify or mitigate the impact of metabolic and environmental influences on glaucoma progression. We have revised the statement to better reflect the synergistic effects of genetics and metabolism rather than considering environmental factors in isolation.

Revised sentence for inclusion in the introduction of the manuscript: "Glaucoma risk is influenced by both genetic and metabolic factors, with emerging evidence suggesting that gene-environment interactions may play a greater role in conferring disease risk than independent exposures alone.” (Lines 95-97)

(7) Lines 156-161: In model 4, rather than stating that the very small increase in AUC with the addition of metabolic data compared to clinical and genetic data alone, "modestly enhances the prediction of glaucoma", it may be better interpreted as a marginal difference that was statistically significant due to the very large sample size but not clinically significant.

Thank you for your suggested comment.

We have adjusted the wording by changing “modestly” to “marginally” to address that the statistical significance is in the context of the study’s large sample size in the results section (Line 162) and throughout the manuscript.

NB: We made other minor edits to correct minor grammatical errors, improve clarity, and streamline the revised manuscript. Furthermore, the paragraph regarding slit lamp examination in the Methods was inadvertently omitted but is added back in the revised manuscript (Lines 571-579).

References:

(1) Kim J, Kang JH, Wiggs JL, et al. Does Age Modify the Relation Between Genetic Predisposition to Glaucoma and Various Glaucoma Traits in the UK Biobank? Invest Ophthalmol Vis Sci. 2025;66(2):57. doi:10.1167/iovs.66.2.57

(2) Cenedella RJ. Lipoproteins and lipids in cow and human aqueous humor. Biochim Biophys Acta BBA - Lipids Lipid Metab. 1984;793(3):448-454. doi:10.1016/0005-2760(84)90262-5

(3) Minassian DC, Reidy A, Coffey M, Minassian A. Utility of predictive equations for estimating the prevalence and incidence of primary open angle glaucoma in the UK. Br J Ophthalmol. 2000;84(10):1159-1161. doi:10.1136/bjo.84.10.1159

(4) Pieri K, Trichia E, Neville MJ, et al. Polygenic risk in Type III hyperlipidaemia and risk of cardiovascular disease: An epidemiological study in UK Biobank and Oxford Biobank. Int J Cardiol. 2023;373:72-78. doi:10.1016/j.ijcard.2022.11.024

(5) Zhao H, Pasquale LR, Zebardast N, et al. Screening by glaucoma polygenic risk score to identify primary open-angle glaucoma in two biobanks: An updated report. ARVO 2025 meeting. Published online 2025.

(6) Zilberter Y, Gubkina O, Ivanov AI. A unique array of neuroprotective effects of pyruvate in neuropathology. Front Neurosci. 2015;9. doi:10.3389/fnins.2015.00017

(7) Quansah E, Peelaerts W, Langston JW, Simon DK, Colca J, Brundin P. Targeting energy metabolism via the mitochondrial pyruvate carrier as a novel approach to attenuate neurodegeneration. Mol Neurodegener. 2018;13(1):28. doi:10.1186/s13024-018-0260-x

(8) Gray LR, Tompkins SC, Taylor EB. Regulation of pyruvate metabolism and human disease. Cell Mol Life Sci. 2014;71(14):2577-2604. doi:10.1007/s00018-013-1539-2

(9) Harder JM, Guymer C, Wood JPM, et al. Disturbed glucose and pyruvate metabolism in glaucoma with neuroprotection by pyruvate or rapamycin. Proc Natl Acad Sci. 2020;117(52):33619-33627. doi:10.1073/pnas.2014213117

(10) Kim MJ, Lee H, Chanda D, et al. The Role of Pyruvate Metabolism in Mitochondrial Quality Control and Inflammation. Mol Cells. 2023;46(5):259-267. doi:10.14348/molcells.2023.2128

(11) Wang X, Perez E, Liu R, Yan LJ, Mallet RT, Yang SH. Pyruvate Protects Mitochondria from Oxidative Stress in Human Neuroblastoma SK-N-SH Cells. Brain Res. 2007;1132(1):1-9. doi:10.1016/j.brainres.2006.11.032

(12) Tilton WM, Seaman C, Carriero D, Piomelli S. Regulation of glycolysis in the erythrocyte: role of the lactate/pyruvate and NAD/NADH ratios. J Lab Clin Med. 1991;118(2):146-152.

(13) Li X, Yang Y, Zhang B, et al. Lactate metabolism in human health and disease. Signal Transduct Target Ther. 2022;7(1):305. doi:10.1038/s41392-022-01151-3

(14) Zhang ZQ, Xie Z, Chen SY, Zhang X. Mitochondrial dysfunction in glaucomatous degeneration. Int J Ophthalmol. 2023;16(5):811-823. doi:10.18240/ijo.2023.05.20

(15) Ju WK, Perkins GA, Kim KY, Bastola T, Choi WY, Choi SH. Glaucomatous optic neuropathy: Mitochondrial dynamics, dysfunction and protection in retinal ganglion cells. Prog Retin Eye Res. 2023;95:101136. doi:10.1016/j.preteyeres.2022.101136

(16) Jassim AH, Inman DM, Mitchell CH. Crosstalk Between Dysfunctional Mitochondria and Inflammation in Glaucomatous Neurodegeneration. Front Pharmacol. 2021;12. doi:10.3389/fphar.2021.699623

(17) Yang TH, Kang EYC, Lin PH, et al. Mitochondria in Retinal Ganglion Cells: Unraveling the Metabolic Nexus and Oxidative Stress. Int J Mol Sci. 2024;25(16):8626. doi:10.3390/ijms25168626